# Enteric glial S100B controls rhythmic colonic functions by regulating excitability and specificity in gut motor neurocircuits

Beatriz Thomasi [iD], Rafaella Lavalle, Jonathon L. McClain, Julia Jamka [iD], Luisa Seguella and Brian D. Gulbransen [iD]

*Department of Physiology, Michigan State University, East Lansing, MI, USA*

Handling Editors: Kim Barrett & Bernard Drumm

The peer review history is available in the Supporting information section of this article (https://doi.org/10.1113/JP289410#support-information-section).

**Abstract figure legend** During colonic motor complex activity, enteric glia play a pivotal role by releasing the S100 calcium-binding protein B (S100B). This gliotransmitter has both intra- and extracellular functions, including the regulation of neuronal and glial excitability within the enteric nervous system. By manipulating S100B signalling, we found that glial S100B controls neuronal rhythmic behaviours, a process that involves extracellular $Ca^{2+}$ regulation. Moreover, the release of S100B shapes functional interactions within cholinergic myenteric neurocircuits, an essential component of the enteric central pattern generator.

**Abstract** Patterns of gut motility, such as colonic motor complexes, are controlled by central pattern generators (CPG) in the enteric nervous system; however, the mechanisms that co-ordinate enteric neural networks underlying this behaviour remain unclear. Evidence from similar CPGs in the brain suggests that glia play key roles through mechanisms involving the S100 calcium-binding protein B (S100B). Enteric glia are abundant in enteric neural networks and engage in bi-directional

**Beatriz Thomasi** is a postdoctoral researcher at Michigan State University, in the Gulbransen lab. She obtained her bachelor's degree in Biological Sciences (2015) and her PhD in Neuroscience (2021) at Universidade Federal Fluminense, Brazil. Her research focuses on the role of enteric glia in gut physiology, inflammation and Parkinson's disease, particularly using mouse transgenic lines and preclinical models. She is interested in translational aspects of enteric neuroscience, neuro-glial communication mechanisms and the modulation of gliotransmission in health and disease.

interactions with neurons, but whether enteric glia shape enteric CPG behaviours through similar mechanisms remains unclear. Here, we show that S100B release by myenteric glia is necessary to sustain colonic motor complex behaviour in the gut. Calcium imaging experiments in whole mounts of myenteric plexus from *Wnt1*[Cre2GCaMP5g-tdTom] mice revealed that the effects of manipulating S100B using selective drugs are a result of changes in neuron and glial activity in myenteric neurocircuits. S100B exerts major regulatory effects over cholinergic neurons, which are considered essential for colonic motor complex initiation and control, and recordings in samples from *ChAT*[CreGCaMP5g-tdTom] mice showed that S100B regulates spontaneous activity among cholinergic neurons and their interactions with other neurons in myenteric networks. These results extend the concept of glia in CPGs to the gut by showing that enteric glial S100B is a critical regulator of rhythmic gut motor function that acts by modulating glial excitability, neuronal behaviours and functional connectivity among neurons. A deeper understanding of this previously unknown glial regulatory mechanism could, therefore, be important for advancing therapies for common gastrointestinal diseases.

(Received 4 June 2025; accepted after revision 13 August 2025; first published online 13 September 2025)

**Corresponding author** B. D. Gulbransen: Department of Physiology, Michigan State University, 567 Wilson Road, East Lansing, MI 48824, USA. Email: gulbrans@msu.edu

**Key points**

- Patterns of gut motility such as colonic motor complexes (CMC) are considered to be controlled by central pattern generators housed in the myenteric plexus of the enteric nervous system.
- Brain central pattern generators studies suggest that glia play key roles through mechanisms involving the protein S100 calcium-binding protein B (S100B).
- This work identifies enteric glial S100B as a regulator of enteric glial and neuronal excitability, through mechanisms of $Ca^{2+}$ regulation that are independent of the RAGE (i.e. receptor for advanced glycation end-products) signalling pathway.
- Enteric glial S100B also controls cholinergic neuronal rhythmic behaviours and functional interactions inside enteric excitatory neurocircuits.
- Our data suggests a novel mechanism by which enteric glia control patterns of gut motor activity through actions of S100B. These observations provide major new insight into mechanisms that regulate fundamental patterns of gut motility and suggest that changes in S100B may be important for understanding changes in gut physiology that occur following disease.

# Introduction

Rhythmic motor behaviours underlie many of the most fundamental biological processes experienced in daily life such as respiration, locomotion, mastication, swallowing and vocalization (Calabrese & Marder, 2025). Such patterns of motor activity are produced by discrete neural networks referred to as central pattern generators (CPG). CPGs autonomously generate rhythmically timed neuronal bursting activity that results in patterns of motor activity in the absence of peripheral sensory input. Rhythmic neuronal bursting is the product of several factors including the properties of the individual cells, synaptic connections among the ensemble of neurons and chemical neuromodulators that overly the circuit. The basic pattern produced by a CPG is also modified by inputs from other neural centres and environmental cues

to adjust motor behaviour appropriately to the conditions. Despite the importance of CPGs in controlling core physiological processes, mechanisms that control CPG initiation, cessation, pattern and timing remain poorly understood in many cases.

There is a growing consensus that gut motor programs such as the colonic motor complex (CMC) are controlled by CPGs contained within the enteric nervous system (ENS) (Spencer, Costa et al., 2021; Wood, 2008). The ENS is the largest and most complex division of the autonomic nervous system and displays integrative capacities that are more 'brain-like' than other divisions of the peripheral nervous system. CMCs are entirely under ENS command, occur independent of sensory stimuli and display alternating rhythmic motor activity typical of CPGs (Broadhead & Miles, 2021; Spencer, Costa et al., 2021). Multiple mechanisms have been proposed to

explain how gut CMC behaviour is generated, including potential subtypes of pacemaker neurons (Smith & Koh, 2017), cyclical release of tonic neural inhibition of interstitial cells of Cajal (Koh et al., 2022) and co-ordinated activity among large areas of interneurons that occurs as an emergent network property (Spencer, Travis et al., 2021). However, no pacemaker subtype of neuron has been identified, and while co-ordinated activity among excitatory interneurons and release of tonic inhibition are key to information flow, they do not explain how CMCs are generated, patterned or modulated.

An emerging concept is that glial cells play a central role in orchestrating CPG behaviours (Broadhead & Miles, 2021; Turk et al., 2022). Astrocytes are integral in regulating the activities of neural circuits and control key aspects needed to sustain CPG activity, such as synaptic transmission, ionic balance and energy supply (Montalant et al., 2021). Blocking mechanisms of astrocytic communication or uncoupling the astrocytic syncytium is sufficient to disrupt, if not entirely halt, rhythmic neuronal bursting in CPGs (Condamine et al., 2018). This effect appears to be mediated by interrupting processes by which astrocytes monitor neuron activity and release transmitters referred to as 'gliotransmitters' that adjust the activity of the surrounding neurons to optimize performance in various environmental conditions (Gourine et al., 2010; Witts et al., 2012). One gliotransmitter that has gained particular attention in CPG behaviours is S100 calcium-binding protein B (S100B). S100B is a $Ca^{2+}$ binding protein that is mainly expressed by glia in the nervous system and exerts control over both intra- and extracellular functions (Donato et al., 2009a). Intracellular S100B transmits changes in cytoplasmic $Ca^{2+}$ into cellular responses involved in glial cell proliferation, migration and differentiation, whereas external S100B exerts diverse actions by buffering extracellular $Ca^{2+}$ and by engaging receptors for advanced glycation end-products (RAGE). S100B is secreted by astrocytes in response to neural activity through mechanisms that involve connexin 43 hemichannels (Condamine et al., 2018) and acts to regulate patterned neuronal bursting through actions on extracellular $Ca^{2+}$ homeostasis (Morquette et al., 2015a). This mechanism plays an important role in controlling rhythmogenesis in CPGs located in the dorsal part of the trigeminal main sensory nucleus (Condamine et al., 2018; Morquette et al., 2015a) but whether similar mechanisms might govern CPG behaviour in the gut is unknown.

Enteric glia are unique neuroglia of the ENS and the only cellular source of S100B within gut motor neurocircuits. Cross-talk between enteric neurons and glia is extensive and glial signalling mechanisms enacted in response to synaptic transmission function to modulate enteric neural circuitry underlying gut motility (Ahmadzai et al., 2021a; Boesmans, Cirillo et al., 2013; Boesmans, Martens et al., 2013; Delvalle, Dharshika et al., 2018; Gulbransen & Sharkey, 2009a; Seguella et al., 2022). Moreover, enteric glial $Ca^{2+}$ activity increases during the occurrence of CMCs and occurs in association with $Ca^{2+}$ activity in neurons and varicosities (Broadhead et al., 2012). Yet, whether enteric glial signalling processes control enteric CPG behaviours as astrocytes do in the brain remains unknown. Given the similar functions of enteric glia and astrocytes in neural networks, the occurrence of neuron-glial cross-talk during CMCs (Broadhead et al., 2012) and that enteric glia are rich in S100B, we reasoned that S100B is a probable candidate by which enteric glia modulate the behaviour of enteric CPG neurocircuits underlying CMC behaviours in the gut. To address this question, we tested the effects of manipulating intracellular and extracellular S100B on neural and glial activity in enteric CPG neurocircuits using transgenic mice that express calcium sensors in enteric neurons, glia or only cholinergic enteric neurons, and studied the effects on CMC behaviour in organ bath experiments. The data show a striking dependence of CMC behaviour on S100B that stems from profound effects that it exerts on glial and neuronal excitability and rhythmicity among colonic neurons. These observations suggest a novel mechanism by which enteric glia control patterns of gut motor activity through actions of S100B.

## Methods

### Ethical approval

All experimental procedures on human samples were approved by the Ethics Committee of St. Orsola-Malpighi Hospital for handling and analysis of tissue samples from patients with severe gut dysmotility (EM/146/2014/O). All aspects of the human studies were conducted in accordance with the principles outlined in the *Declaration of Helsinki*. Written informed consents were obtained before tissue collection. This study did not involve any intervention or prospective recruitment and was therefore not subject to clinical trial registration (*Declaration of Helsinki*, paragraph 35).

All animal work was conducted following the National Institutes of Health (NIH) Guide for the Care and Use of Laboratory Animals and received approval from the Institutional Animal Care and Use Committee (IACUC) at Michigan State University (MSU) PROTO202400108. Mice were housed in a temperature-controlled environment under a 12:12 h light/dark photocycle with unrestricted access to food (Diet Number 2919; Envigo, Indianapolis, IN, USA) and water. The investigators understand the ethical principles under which the journal operates, and this work complies with the animal ethics checklist endorsed by the ARRIVE

guidelines 2.0 adopted by *The Journal of Physiology* (Sert et al., 2020).

## Human colon sample collection from healthy subjects

Full-thickness colon tissue samples were obtained from four adult control patients (two females, two males; aged 48–73 years) undergoing surgery for non-complicated gastrointestinal tumours. Immediately after resection, tissues were fixed in cold neutral 4% formaldehyde (Kaltek, Veneto, Italy), paraffin-embedded, sectioned at 5 μm thickness, and mounted on poly-L-lysine-coated slides (Thermo Fisher Scientific, Waltham, MA, USA) for further analysis.

## Animals

Transgenic mice expressing the genetically encoded calcium indicator GCaMP5g in enteric neurons and glia were bred in-house. *Wnt1*[Cre2GCaMP5g-tdTom] and *Chat*[CreGCaMP5g-tdTom] mice were generated by crossing B6.Cg-E2f1Tg(Wnt1-cre)2Sor/J mice (The Jackson Laboratory, Bar Harbor, ME, USA; stock no. 02 2501; RRID: IMSR_JAX:02 2501) (Ahmadzai et al., 2021a) or *Chat*-IRES-Cre mice (The Jackson Laboratory; stock no. 03 1661; RRID: IMSR_JAX:03 1661) (Rossi et al., 2011), respectively, with Polr2atm1(CAG-GCaMP5g, tdTomato)Tvrd (RRID: IMSR_JAX:02 4477) mice. Experiments were conducted with male and female mice, 8–12 weeks old weighing 25–30 g. In line with the journal ethical guidelines, mice weighing 25–30 g were killed without prior administration of anaesthesia. Animals were killed by trained personnel using cervical dislocation with a second step of decapitation. Animal death was confirmed by observation of heartbeat cessation.

## Colonic motor complex recordings

CMCs were assessed in organ bath experiments using isolated colons from Polr2atm1 mice as described previously (McClain et al., 2015a). Whole intact colons were collected in 30–32°C Dulbecco's modified Eagle's medium/F-12 (11039; Gibco, Waltham, MA, USA) and luminal contents were gently flushed. Colons were mounted on a stainless-steel rod and the oral and aboral ends were secured with surgical silk. Force transducers (Grass Instruments, Quincy, MA, USA) were attached to the gut wall via surgical silk ∼2 cm apart. Tissue was immersed in a bath containing Dulbecco's modified Eagle's medium/F12 media maintained at 37°C with an initial tension of 0.5 g. Spontaneous CMC activity was recorded in LabChart 8 (ADInstruments, Colorado Springs, CO, USA) over 20 min before initiating experiments. CMCs were defined as contractions originating at the oral site and propagating to the aboral site. The last 5 min of this acclimatization period was used as a baseline for analysis. Drugs were bath applied for 30 min and the resultant changes to CMC amplitude and integral were quantified in relation to baseline.

## Longitudinal muscle myenteric plexus (LMMP) whole-mount preparations

Colons were carefully extracted and kept in ice-cold Krebs solution. LMMP preparations were isolated from colons using the rod dissection method (Smith et al., 2013) to perform S100B release assays. Mesenteric border fat was removed and the resulting LMMPs were rinsed with Krebs (2 × 5 min) for assay preparation.

For immunohistochemistry reactions, the intact colons were opened along the mesenteric border and pinned flat with mucosa layer facing up in Sylgard-coated Petri dishes (Dow Corning, Midland, MI, USA). Tissues were fixed in 4% paraformaldehyde for 2 h at room temperature and washed 3 times with phosphate-buffered saline (PBS). Mucosa, submucosa and circular muscle layers were carefully dissected to obtain a fixed LMMP.

## Enzyme-linked immunosorbent assay (ELISA)

Following rod dissection and washing with Krebs buffer, LMMPs were incubated for another 30 min in a 48 well plate in fresh Krebs or Krebs containing arundic acid (AA) (50 μM; Cayman Chemicals, Ann Arbor, MI, USA). Media was collected and centrifuged at 15142 g-force at 4°C. Supernatants were collected and stored at –80°C until processing. LMMP samples were weighed, homogenized in liquid nitrogen and resuspended in a buffer containing RIPA buffer, phosphatase inhibitors and distilled water before being centrifuged at 15142 g-force (4°C) followed by collection of the supernatant. S100B content was measured via an ELISA using a Mouse S100B/S100 Beta ELISA Kit – sandwich (LSBio, Inc., Seattle, WA, USA). Briefly, supernatants and LMMPs samples were applied to pre-coated kit wells and incubated for 1 h at 37°C. Biotin-conjugated detection antibody was added and incubated for 1 h at 37°C. Unbounded antibody was washed and an avidin-horseradish peroxidase conjugate to bind biotin was added for 30 min. Unbound avidin-horseradish peroxidase was washed and 3,3′,5,5′-tetramethylbenzidine substrate was added to the wells for 10–20 min at 37°C. Colour development was monitored until an optimal signal was achieved and then stopped by application of sulphur of sulphuric acid solution. Optical density was read using a plate reader at 450 nm and measurements were fitted to a standard curve to determine the S100B concentration (pg mL$^{-1}$).

**Table 1. Primary and secondary antibodies used for immunohistochemistry in this work and detailed information.**

| Antibody target | Origin | Vendor | #catalog | RRID | Concentration |
|---|---|---|---|---|---|
| Primary antibodies | | | | | |
| Peripherin | Mouse | SantaCruz Biotechnology | Sc-377 093 | AB_2 923 264 | 1:200 |
| S100B | Rabbit | Abcam | ab52642 | AB_882426 | 1:200 |
| S100B | Chicken | SynapticSystems | 287006 | AB_2713986 | 1:500 |
| GFAP | Chicken | Abcam | ab4674 | AB_304558 | 1:200 |
| PGP9.5 | Guinea pig | Neuromics | GP14104 | AB_2210625 | 1:500 |
| RAGE | Rabbit | Abcam | ab37647 | AB_777613 | 1:200 |
| Secondary antibodies | | | | | |
| Alexa Fluor 488 | Donkey anti-chicken | Jackson Laboratories | 703-545-155 | AB_2340375 | 1:400 |
| Alexa Fluor 488 | Donkey anti-mouse | Jackson Laboratories | 715-545-150 | AB_2340846 | 1:400 |
| Alexa Fluor 488 | Donkey anti-guinea pig | Jackson Laboratories | 706-545-148 | AB_2340472 | 1:400 |
| Alexa Fluor 594 | Donkey anti-rabbit | Jackson Laboratories | 711-585-152 | AB_2340621 | 1:400 |
| Alexa Fluor 647 | Donkey anti-rabbit | Jackson Laboratories | 711-605-152 | AB_2492288 | 1:400 |

Final values were corrected by protein concentration determined by BCA protein assay (Thermo Fisher Scientific).

### Immunohistochemistry

LMMP preparations were permeabilized with PBS containing 0.1% Triton X-100 (3 × 10 min) and then incubated for 45 min in a blocking solution containing 4% normal goat serum, 0.4% Triton X-100 and 1% bovine serum albumin. Following the blocking step, the preparations were incubated with primary antibodies overnight (Table 1). On the second day, the samples were washed in PBS and incubated with secondary antibodies (Table 1). After final washes, LMMP samples were mounted using DAPI-Fluoromount G (Southern Biotec, Birmingham, AL, USA). Each reaction had negative controls without application of primary antibodies.

### Fluorescence labelling imaging

Immunofluorescence labelling was observed using a 20× objective lenses (N.A. 0.75, Plan Fluor; Nikon, Tokyo, Japan) of an upright epifluorescence microscope (Nikon Eclipse Ni) with a Retiga 2000R camera (Qimaging, Teledyne Surrey, BC, Canada) controlled by MetaMorph, version 7.10.1.161 (Molecular Devices, San Jose, CA, USA). Representative images were acquired using a LSM880 NLO microscope (Zeiss, Oberkochen, Germany) (20× water immersion, N.A. 1.0) using ZEN 2.3 software (Zeiss) and a Stellaris 5 (40× oil immersion HC PLAPO CS2 objective, n.a 1.30) (Leica, Wetzlar, Germany) with a tunable white light laser and variable slit detection with HyD detectors controlled by LAS X version 4.6.0.27096 (Leica). Acquisition parameters of fluorescence intensity were optimized to each labelling.

### Calcium imaging of spontaneous activity in the myenteric plexus

Colons from $Wnt1^{Cre2GCaMP5g-tdTom}$ and $ChAT^{CreGCaMP5g-tdTom}$ mice were extracted and kept in ice-cold Krebs solution. Live cell imaging was conducted using whole mounts of circular muscle-myenteric plexus (CMMP) preparations as previously described (Ahmadzai et al., 2021a; Morales-Soto et al., 2024). Briefly, the mucosa and longitudinal muscle were removed by microdissection to reveal the myenteric plexus attached to circular smooth muscle. Fluorescence imaging was conducted using an upright BX51WI fixed-stage microscope (Olympus, Tokyo, Japan) with wide-field water-immersion 20× objective lens (XLUMPLFLN20xW, 1.0 numerical aperture; Olympus). Fluorescence excitation was supplied by a Lumencor AURA III light engine (Lumencor, Beaverton, OR, USA). To excite GCaMP5g, light was passed through a 485/20 nm bandpass filter and filtered emission with a 515 nm longpass filter. For the tdTomato channel, CMMPs were excited by passing the light through a 535/20 nm bandpass filter and filtered through a 610/75 nm bandpass emission filter before detection. Neuron and glial activity were simultaneously imaged in samples from $Wnt1^{Cre2GCaMP5g-tdTom}$ mice that express

GCaMP in both enteric glia and neurons (Wnt1+ cells), whereas cholinergic neuron activity was specifically recorded using samples from $Chat^{CreGCaMP5g\text{-}tdTom}$ mice. Fresh, prewarmed (37°C) Krebs solution was continuously applied at a flow rate of 2–3 mL min$^{-1}$. Spontaneous Ca$^{2+}$ activity in enteric neurons, glia and ChAT(+) neurons were recorded at a frame rate of 2 Hz for 3–5 min. Time-lapses and microphotographs were acquired by a Photometrics Prime BSI camera controlled by MetaMorph (Molecular Devices) or NIS Elements AR, version 6.10.01 (Nikon).

## Solutions and drugs

Modified Krebs buffer, containing (in mм): 121 NaCl, 5.9 KCl, 2.5 CaCl$_2$, 1.2 MgCl$_2$, 1.2 NaH$_2$PO$_4$, 10 HEPES, 21.2 NaHCO$_3$, 1 pyruvic acid and 8 glucose (pH adjusted to 7.4 with NaOH) was used to dissect, incubate and superfuse samples in all live imaging experiments. Arundic acid (dissolved in ethanol; Cayman Chemicals) was further diluted in Krebs buffer to a final working concentration of 50 μм for Ca$^{2+}$ imaging experiments and 300 μм for CMC recordings. Pentamidine isethionate salt (Sigma, St Louis, MO, USA) was prepared as a 500 μм stock solution and diluted to a final working concentration of 10 μм for a 1 h incubation of wholemounts and 30 μм for CMC. Anti-S100B antibodies (EP1576Y) (Abcam, Cambridge, UK) were used at a 1:1000 dilution in Krebs buffer during Ca$^{2+}$ imaging experiments and 1:1000 and 1:500 during CMC recordings. FPS-ZM1 (Tocris, Minneapolis, MN, USA) was applied at 1 μм to test effect of RAGE antagonism during CMCs. The hemichannel blocker Gap26, targeting Connexin 43 (43Gap26; Anaspec, Fremont, CA, USA), was diluted in distilled water and used at a working concentration of 100 μм in wholemounts for a 30 min incubation. $Wnt1^{Cre2GCaMP5g\text{-}tdTom}$ wholemounts were incubated for 30 min to 1 h prior to imaging. $ChAT^{CreGCaMP5g\text{-}tdTom}$ wholemounts underwent the following sequential treatments: spontaneous recording in Krebs buffer, incubation with arundic acid (50 μм, 20 min), washout (15 min) and incubation with recombinant S100B protein (50 μg mL$^{-1}$; Novus Bio, Centennial, CO, USA) to verify temporal responses from the same ganglia.

## Data analysis

For immunoassays, each animal was considered as one *n*. For Ca$^{2+}$ imaging quantification, recording processing and region of interest (ROI) selection were performed as previously described (Ahmadzai et al., 2021a). Briefly, image sequences from Ca$^{2+}$ imaging experiments were background-subtracted and aligned using Fiji software plugins (National Institutes of Health, Bethesda, MD, USA). tdTomato expression and neuronal morphology were used to create ROIs corresponding to individual cells. The mean grey value of each ROI was calculated in Fiji, and relative changes in fluorescence ($\Delta F/F_0$) were computed in Excel (Microsoft Corp., Redmond, WA, USA). A cell was considered spontaneously active if the average response subtracted from the maximum intensity exceeded the first quartile value plus three times the standard deviation (SD). For amplitude correction, the first quartile was subtracted from the maximum intensity value to remove noise for each cell. For population analysis, the relative number of responsive cells in each ganglion was counted as an *n*. A minimum of two ganglia per animal was analyzed. Hundreds of neurons and/or glia were pooled for amplitude and frequency analyses and considered individual *n* values. Detailed *n* of mice, ganglia and cells are provided wjere appropriate. Representative Ca$^{2+}$ imaging traces were selected based on the cell response profiles.

Functional connectivity was calculated using Detect software, a MATLAB-based tool for Ca$^{2+}$ imaging analysis (Desai et al., 2024). Pre-processed time-lapse sequences were uploaded to Detect, and ROIs were manually defined. Pairwise correlation coefficients between every pair of neurons were calculated. Pairwise normalization was performed using Power Query by filtering pair-by-pair values and applying the formula: (Correlation – Min_correlation)/(Max_correlation – Min_correlation). Earth Mover's Distance (EMD) was used to measure frequency distributions and oscillatory patterns between each pair of neurons. Normalization was similarly performed using Power Query but with the formula: 1 – (EMD – Min_EMD)/(Max_EMD – Min_EMD). Global synchronization was calculated using a weighted average of 70% pairwise correlation and 30% EMD. The distinct metrics were combined here to provide the most accurate and biologically relevant neuronal interactions, considering temporal alignment and pattern similarity (Baruzzi et al., 2023; Xiong et al., 2020). Averaged values from each ganglion were plotted as an *n* for analyses involving the number of clusters based on pairwise correlation coefficients, normalized pairwise correlations and EMD indices. Pairwise raw value plots considered each correlation value between a pair of neurons as an *n*. Detailed *n* of mice, ganglia and cells are found in the captions. Heatmaps are shown from a representative ganglion.

## Statistical analysis

Male and female mice were analyzed separately and combined if no statistical differences were observed. Data was analyzed using Prism, version 10 (GraphPad

Software Inc., San Diego, CA, USA). Shapiro–Wilk and Kolmogorov–Smirnov tests verified data normality. Parametric and non-parametric tests were applied accordingly. Student's *t* test was applied for normal data, while non-Gaussian data were subjected to the Mann–Whitney test. One-way analysis of variance Kruskal–Wallis and Dunn's multiple comparison test was used for three or more groups. Data are presented as the mean ± SD.

## Results

### Enteric glia contain and release S100B in enteric motor neurocircuits

S100B is a $Ca^{2+}$-binding protein that has diverse intra- and extracellular functions (Donato et al., 2009*a*). Among these, regulating cellular excitability and intercellular signalling have emerged as important roles for S100B in CPGs in the CNS (Morquette et al., 2015*a*). Enteric CPGs that control gut motor responses are housed in the myenteric plexus (Sharkey & Mawe, 2022) where S100B is abundant in enteric glia that surround enteric neurons (Fig. 1*A* and *B*). Myenteric glia are 'canonical' enteric glia that develop with enteric neurons and act to optimize ENS functions through bi-directional signalling with neurons (Ahmadzai et al., 2021*b*; McClain et al., 2015*b*; Seguella & Gulbransen, 2021; Windster et al., 2024). Co-labelling with antibodies against glial fibrillary acidic protein (GFAP), a marker robustly expressed by myenteric glia (Jessen & Mirsky, 1980; Windster et al., 2024), and peripherin, a pan-neuronal marker that identifies most enteric neurons (Troy et al., 1990), shows that S100B is restricted to glia in myenteric neurocircuits (Fig. 1*A* and *B*). This observation agrees with the expression observed in S100B-GFP reporter mice (Gulbransen & Sharkey, 2009*b*) and with the pattern of expression observed in multiple other species, including guinea pigs (Gulbransen & Sharkey, 2009*b*; Lavoie et al., 2011) and humans (Ahmadzai et al., 2022*a*). Therefore, abundant S100B contained in enteric glia is an evolutionarily conserved feature of enteric motor neurocircuits.

Prior work showed that astrocytes actively secrete S100B into the extracellular space and multiple stimuli, including neurotransmitters/modulators, can augment astrocytic S100B secretion (Donato et al., 2009*b*). Measurements of S100B in supernatants from isolated myenteric plexus preparations showed that enteric glia also tonically secrete S100B into the extracellular space in unstimulated samples (Fig. 1*C*). Blocking S100B production with the drug arundic acid decreased S100B content in myenteric plexus tissue and supernatant samples ($N = 3$, Krebs 8202 ± 1194 *vs.* AA 2671 ± 1363, $P = 0.0379$) (Fig. 1*D* and *E*). Arundic acid mainly impairs S100B release, with less effect on intracellular S100B content (Cordeiro et al., 2020; Ishiguro et al., 2019; Vizuete et al., 2021). In agreement, despite observing reductions in S100B release by myenteric plexus in medium (–30%) and tissue samples (–68%), cellular S100B expression remained unchanged following arundic acid incubation in our healthy samples (Fig. 1*F*). These data show that S100B is released by enteric glia in myenteric neurocircuits and that arundic acid impairs glial S100B release with minimal notable impact on intracellular content.

To support the translational relevance of murine ENS studies to human physiology, PGP9.5 and S100B were labelled in full thickness human colon cross-sections. The reactions reveal the organization of colon layers and ENS with PGP9.5 marking neuronal cell bodies and fibres, and S100B confined to surrounding enteric glia (Fig. 1*G* and *H*). This spatial organization is consistent with that observed in mice and demonstrates that S100B is confined to enteric glia in the myenteric plexus.

### S100B controls colonic motor complexes

CMCs are rhythmic motor patterns of the colon that are controlled by enteric CPGs in the myenteric plexus (Smith et al., 2014; Spencer & Hu, 2020). Given that S100B controls rhythmogenesis in CPGs (Morquette et al., 2015*b*) and S100B is produced by enteric glia in the neuro-circuits that control CMCs, we speculated that enteric glia may exert control over enteric CPGs via S100B. To test this concept, we studied CMC activity in organ bath experiments by adding selective drugs and antibodies to manipulate intracellular and extracellular functions of S100B.

In agreement with prior work (Delvalle, Fried et al., 2018; McClain et al., 2015*a*; Spencer, Costa et al., 2021), spontaneous CMCs occurred in the isolated mouse colon and propagated in an oral to aboral direction (Fig. 2*A*). CMC activity was almost completely abolished when spontaneous S100B release was impaired by adding arundic acid. Arundic acid decreased the amplitude of oral and aboral contractions by 70% and 84% ($N = 6$ mice; oral: Krebs 100 ± 12.78 *vs.* AA 32 ± 22.87, $P < 0.0001$; aboral: Krebs 100 ± 10.16 *vs.* AA 16.95 ± 34.16, $P < 0.0001$) (Fig. 2*B*), respectively, and the integral of contractions by 82% and 89% ($N = 6$, oral: Krebs ± 25.68 *vs.* AA 18.18 ± 13.61, $P < 0.0001$; aboral: Krebs 100 ± 18.67 *vs.* AA 12.22 ± 16.10, $P < 0.0001$) (Fig. 2*C*). Notably, these effects were reversible, and CMC activity resumed after arundic acid washout and return to normal media (Fig. 2*A*, bottom).

The effects of arundic acid suggest that extracellular S100B plays a major role in sustaining CMC activity in the gut. If true, then other manipulations of extracellular S100B should produce similar effects. In support, scavenging S100B with anti-S100B antibodies (ab)

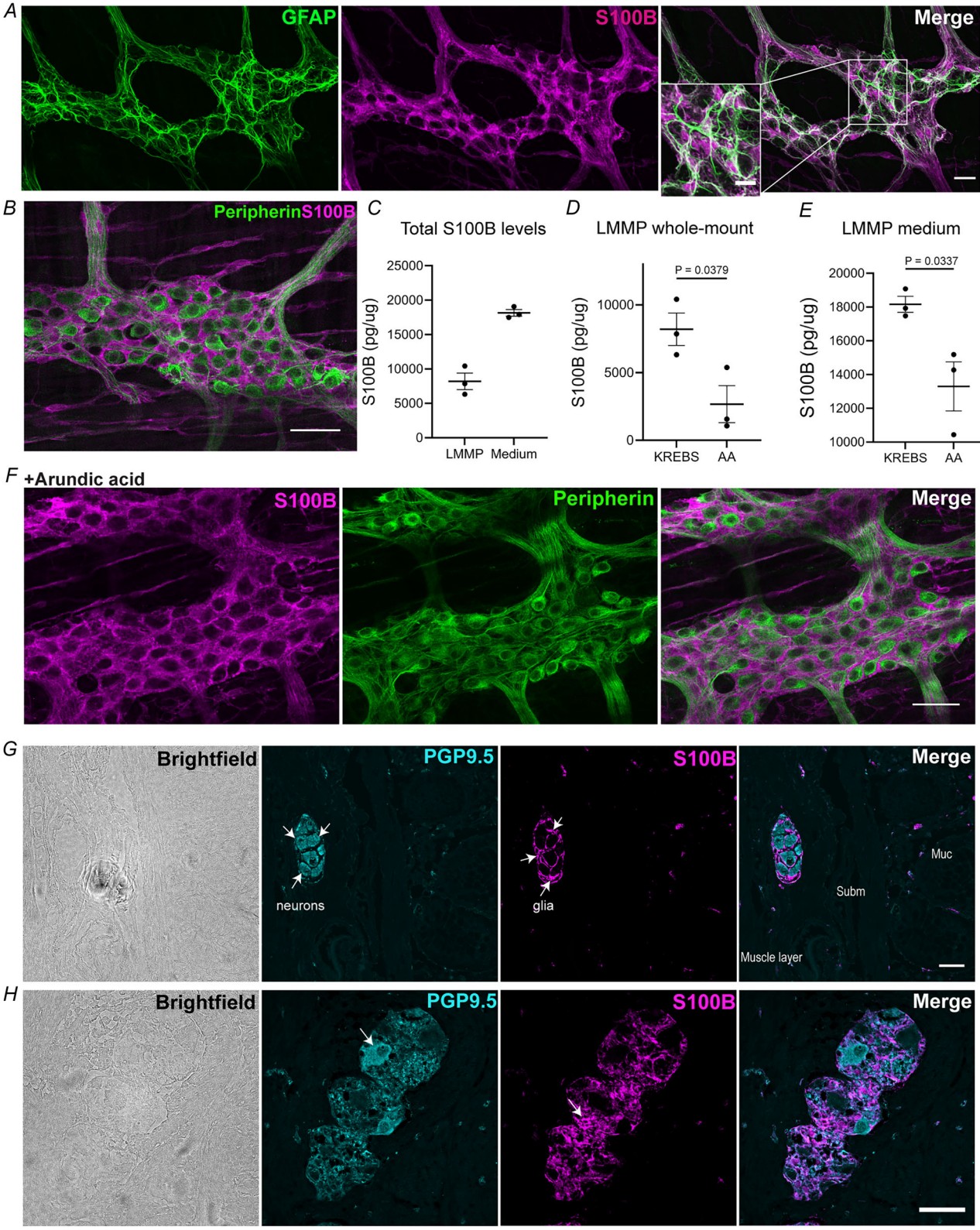

**Figure 1. Enteric glia produce S100B in myenteric neurocircuits**
*A*, representative images of a myenteric ganglion from the mouse colon labelled with antibodies against the glial markers GFAP (green) and S100B (magenta). Note extensive co-labelling between GFAP and S100B in the merged image (right panel) showing that cells containing S100B are myenteric glia. Inset highlights one area of this ganglion at higher magnification. *B*, a representative ganglion co-labelled with antibodies against S100B

(magenta) and the neuronal marker peripherin (green) showing that myenteric neurons do not express S100B. *C*, total S100B levels in longitudinal muscle-myenteric plexus (LMMP) tissue samples from the proximal colon and their respective supernatants (medium). Treatment with the S100B synthesis inhibitor arundic acid (50 μM, 1 h) reduced S100B levels in both tissue (*D*) and supernatant (*E*) ($P = 0.0379$ and $0.0337$, respectively), without affecting cellular markers or colonic cytoarchitecture (*F*). $N = 3$ mice. Data were analyzed using an unpaired Student's *t* test and plotted as the mean ± SD. *A–F*, scale bar = 50 μm. *G*, representative images of brightfield, neuronal marker PGP9.5 (cyan) and glial marker S100B (magenta) in healthy human colon cross-sections. Note the colonic layers and the position of myenteric ganglia between circular and longitudinal smooth muscle layers. *H*, Zoomed images of a myenteric ganglion labeled with markers of neurons and glia showing that S100B labelling is confined to enteric glia as observed in the mouse ENS. *G*, scale bar = 50 μm. *H*, scale bar = 100 μm. Muc = mucosa, Subm = submucosa.

also produced significant deficits in CMC function ($N = 3–6$ mice; oral amplitude: Krebs *vs.* S100B ab 56.24 ± 34.15, $P = 0.0078$; aboral amplitude: Krebs *vs.* S100B ab 62.73 ± 34.16, $P = 0.0494$; oral integral: Krebs *vs.* S100B ab 59.91 ± 41.61, $P = 0.0383$; aboral integral: Krebs *vs.* S100B ab 54.75 ± 47.20, $P = 0.0462$) (Fig. 2*A–D*). S100B mainly exerts its actions in the extracellular space by binding $Ca^{2+}$ or by engaging RAGE (Barger & Eldik, 1992; Hagmeyer et al., 2018; Villarreal et al., 2011; Zimmer & Weber, 2010). Pentamidine is a drug with broad actions on S100B-RAGE axis both by directly disrupting $Ca^{2+}$/p53 binding site of S100B and limiting its ability to bind $Ca^2$ into the glial protein, or by blocking S100B ligand activity on RAGE (Charpentier et al., 2008; Hartman et al., 2013). Doing so by adding pentamidine (30 μM) to the bath solution caused a major failure of CMC activity and decreased the strength of contractions still observed at oral and aboral sites by 82% and 74% in amplitude ($N = 6$ mice; oral: Krebs *vs.* pentamidine (PTM) 19.88 ± 10.82, $P < 0.0001$; aboral: Krebs *vs.* PTM 27.82 ± 19.58, $P = 0.0001$) (Fig. 2*B*), with integrals reduced by 93% and 85%, respectively (oral: Krebs *vs.* PTM 7.26 ± 5.13, $P < 0.0001$; aboral: Krebs *vs.* PTM 16.26 ± 12.54, $P = 0.0002$) (Fig. 2*C*). Additionally, CMC contraction location and propagation directionality were dysregulated following incubation with arundic acid, pentamidine and anti-S100B antibodies (Fig. 2*D*). Together, these observations show that CMCs, a motor behaviour of the colon controlled by an enteric CPG, require S100B through mechanisms that probably involve the extracellular $Ca^{2+}$-binding actions of S100B.

## S100B regulates spontaneous activity among neurons and glia in myenteric neurocircuits

The observation that colonic CMC behaviours require S100B suggests that S100B plays a role in sustaining the activity of enteric neurocircuits that control CMCs. These neurocircuits are housed in the myenteric plexus and composed of enteric neurons and glia that derive from a common pool of *Wnt1+* neural crest progenitors. In prior work, we used this feature to express the genetically calcium indicator GCaMP5g broadly among both enteric neurons and glia to study their activity (Ahmadzai et al.,

2021a, 2022b). We validated *Wnt1*$^{Cre2GCaMP5g-tdTom}$ mice as an effective model to study enteric neuron and glial activity and found that differences in tdTomato expression between enteric neurons and glia facilitate differentiating between the cell types (Fig. 3*A–C*).

Neuronal responses profiles were heterogeneous and included cells that exhibited large peaks, low- and high-amplitude rhythmic peaks and sparse-intermittent responses (Fig. 3*D*), which is consistent with neuronal response profiles observed by others in the mouse myenteric plexus (Debnath et al., 2025). This variety of neuronal behaviours in a single ganglion demonstrates that microcircuits of the ENS are functionally heterogeneous, an inherent feature for CPGs (Gutiérrez-Ibáñez & Wylie, 2024; Marder & Bucher, 2001). Recordings of neurons in the mouse colon myenteric plexus showed that 84% of myenteric neurons are spontaneously active during a period of 2–5 min (Fig. 3*F*). Consistent with prior reports (Ahmadzai et al., 2021a; Dedek et al., 2022), myenteric neurons exhibited larger amplitude $Ca^{2+}$ responses in females than those in males (male 0.1944 ± 0.0127 *vs.* female 0.2844 ± 0.0107, $P < 0.0001$) (Fig. 3*H*).

Astrocytic $Ca^{2+}$ oscillations are an essential component of CPGs in the brain (Broadhead & Miles, 2021; Kadala et al., 2015; Semyanov, 2019; Turk et al., 2022) and recordings of spontaneous enteric glial activity (Fig. 3*C*) within myenteric neural circuits showed that 68% of myenteric glia exhibit spontaneous activity within a recording period of 2 min in enteric CPG networks (Fig. 3*G*). Consistent with prior observations (Ahmadzai et al., 2021b), female glia exhibited larger amplitude $Ca^{2+}$ responses than male glia (male 0.1086 ± 0.0058 *vs.* female 0.1675 ± 0.0094, $P < 0.0001$) (Fig. 3*I*) and glial response profiles displayed patterns of rhythmic, low and high amplitude peaks, similar to spontaneous $Ca^{2+}$ responses observed in astrocytes (Crowe et al., 2024; Gau et al., 2023; Wang et al., 2006) (Fig. 3*E*). Therefore, enteric glia exhibit spontaneous activity and may engage in functional interactions with neighbouring neurons in ways similar to glia in brain CPGs.

Glia control neuronal signalling in CPGs through gliotransmission mechanisms (Pirttimaki et al., 2017) triggered by glial activity encoded by intracellular $Ca^{2+}$ responses (Poskanzer & Yuste, 2011, 2016).

Given that enteric glia are active during CMCs, release S100B into the myenteric plexus and that blocking S100B abolishes CMC function, we hypothesized that S100B acts by regulating spontaneous ENS activity. We began testing our hypothesis by treating CMMPs from $Wnt1^{Cre2GCaMP5g\text{-}tdTom}$ mice with arundic acid (50 μM) to inhibit S100B release and recording the effects on spontaneous neuron and glial activity (Fig. 4*A* and *B*). Similar numbers of neurons and glia exhibited spontaneous activity following incubation with arundic acid (glia: Krebs 68.10 ± 4.8 *vs.* AA 76.04 ± 5.0, $P = 0.266$;

neurons: Krebs 85.45 ± 2.1 *vs.* AA 80.93 ± 3.0, $P = 0.229$) (Fig 4*B*, *C* and *G*) and no differences were observed in the amplitudes of glial (Krebs 0.1324 ± 0.005 *vs.* AA 0.1194 ± 0.004, $P = 0.854$) (Fig. 4*D*) and neuronal (Krebs 0.2457 ± 0.007 *vs.* AA 0.2866 ± 0.010, $P = 0.527$) (Fig. 4*H*) responses when sexes were combined. Given that prior evidence showed that effects of S100B are often sexually dimorphic (Diehl et al., 2007; Jhun et al., 2020; Yang et al., 2008), we considered whether combining both sexes obscured potential sex-specific effects. Stratifying cell responses by sex revealed significant effects of arundic

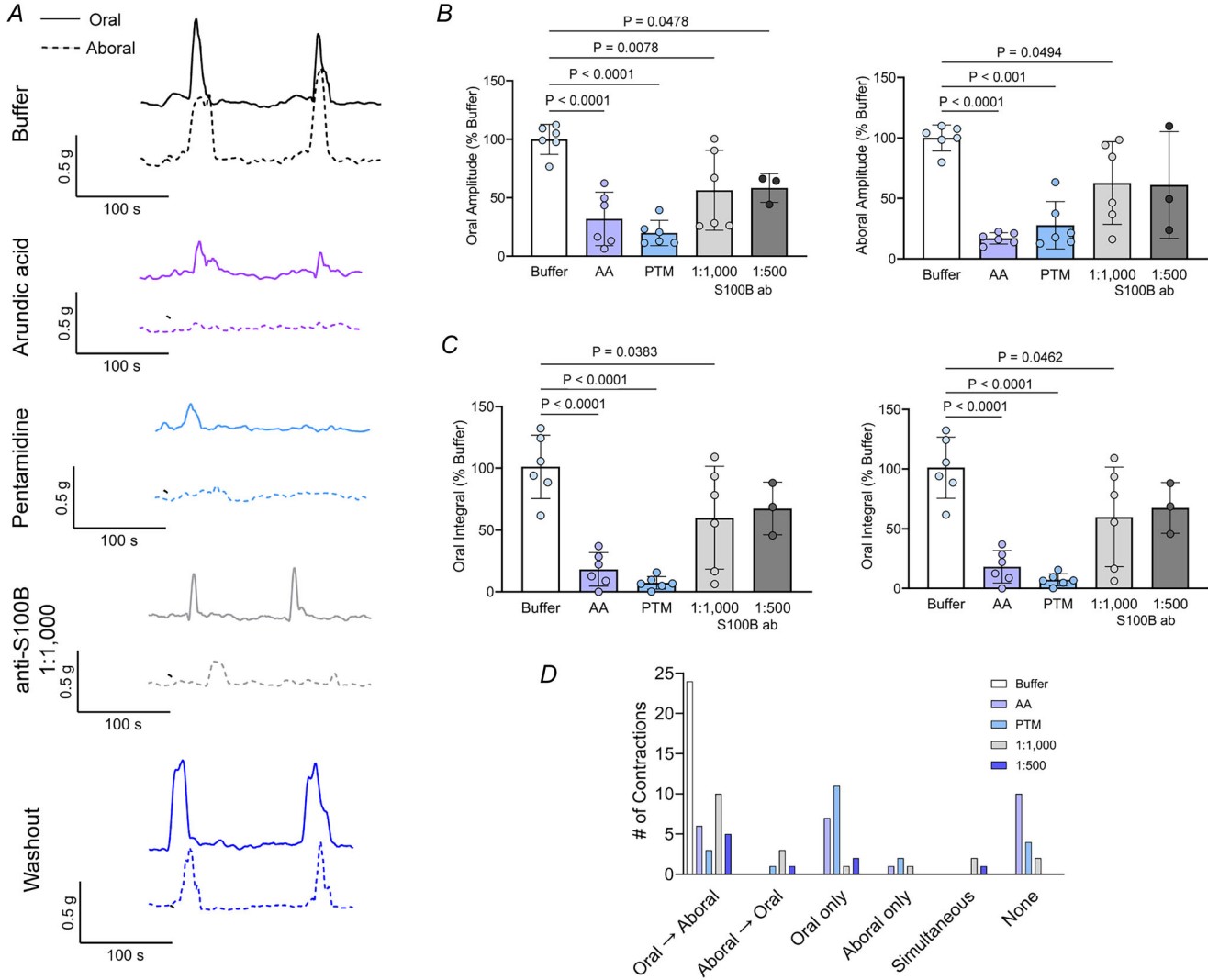

**Figure 2. S100B controls colonic motor complexes**
*A*, representative traces of colonic motor complex (CMC) behaviour in the mouse colon under control conditions and following treatment with drugs to manipulate S100B. Solid lines show motor responses at oral recording sites, whereas dashed lines show motor corresponding responses at aboral recording sites in the same organ. The bottom trace shows CMC activity following arundic acid washout. Quantification of (*B*) CMC contraction amplitude at oral (*P* values = 0.0494, 0.0078 and *P* <0.0001) and aboral sites (*P* = 0.0494, *P* < 0.001 and *P* < 0.0001) and (*C*) integrated contraction area at oral (*P* = 0.0383, *P* < 0.0001) and aboral sites (*P* = 0.0462, *P* < 0.0001) related to buffer condition. *D*, effects of manipulating S100B on CMC propagation. *N* = 3–6 mice. Data were analyzed using one-way analysis of variance followed by a *post hoc* Bonferroni multiple comparisons test. AA = arundic acid, PTM = pentamidine, S100B ab = S100B antibodies.

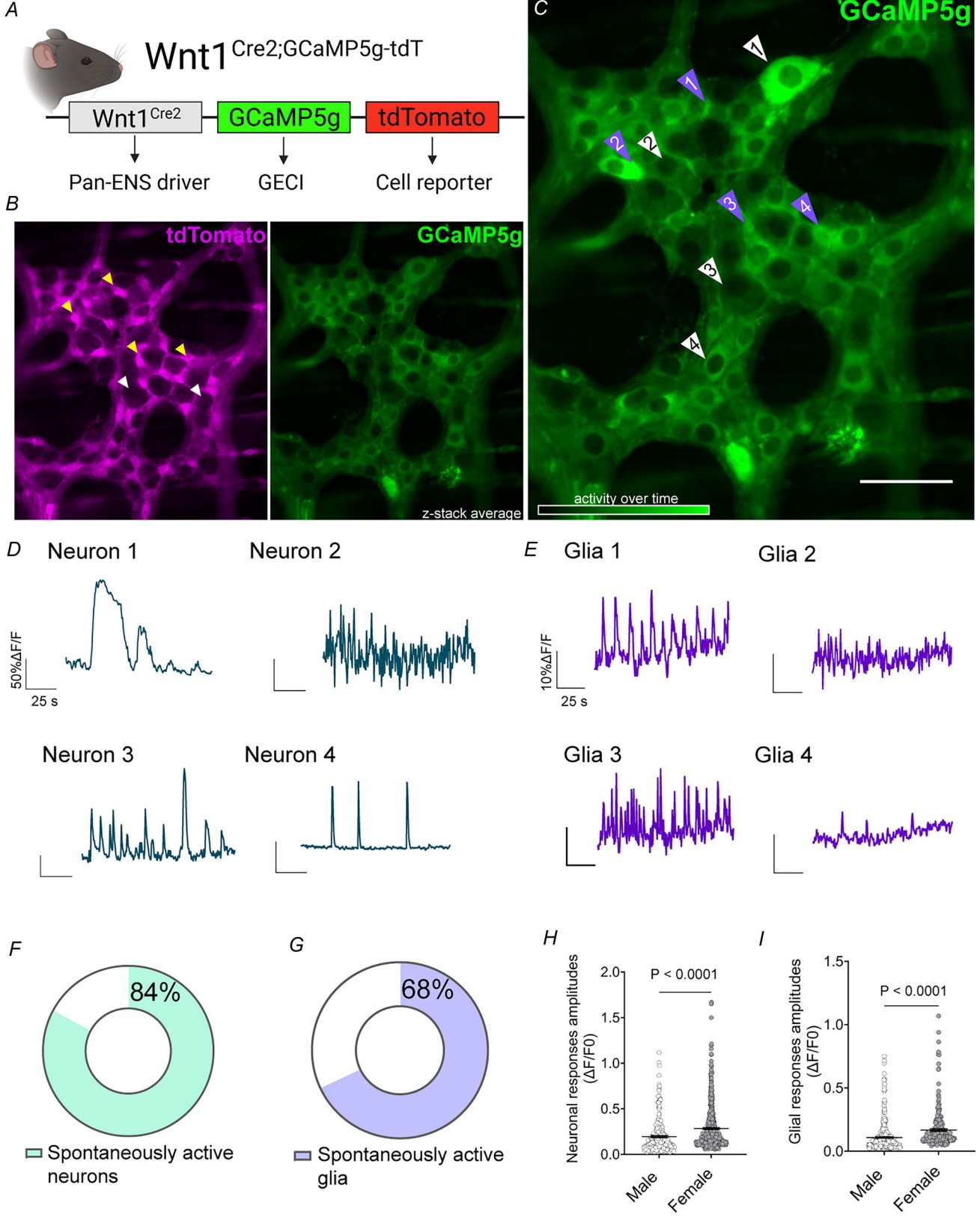

**Figure 3. Spontaneous activity among myenteric glia and neurons in the mouse colon**
*A*, *Wnt1*[Cre2GCaMP5g-tdTom] transgenic mice express the genetically encoded calcium indicator GCaMP5g in myenteric neurons and glia (Wnt1[+] cells). *B*, representative images of tdTomato (magenta, left) and GCaMP5g

(green, right) expression in a myenteric ganglion showing that tdTomato expression is more robust in enteric glia (yellow arrowheads) than neurons (white arrowheads) which, when combined with morphology, enables cell-type-specific identification. C, temporal projection of average spontaneous GCaMP5g activity in this ganglion over 120 s. White arrows point to several representative neurons (1–4) and purple arrows highlight several glia (1–4). Traces of these cells are shown in (D) and (E), respectively. Total percentages of spontaneously active myenteric neurons (F) and glia (G) from all experiments as well as their calcium response amplitudes ($\Delta F/F_0$) ($P < 0.0001$) (H and I). N = 10–16 ganglia from three to five animals; n = 300–500 neurons or glia. Data were analyzed using Student's t test, followed by the Mann–Whitney *post hoc* test. Scale bar = 50 μm.

acid on both neurons and glia. Glia exhibited lower activity following arundic acid in both males and females; however, this effect was most pronounced in females where glial amplitudes were reduced by 22% (Krebs $0.1675 \pm 0.009$ vs. AA $0.1299 \pm 0.006$, $P < 0.0001$) (Fig. 4E and F). By contrast, response amplitudes among neurons from female mice were 15% larger following S100B inhibition (Krebs $0.2844 \pm 0.010$ vs. AA $0.3266 \pm 0.015$, $P = 0.0014$) (Fig. 4I and J), whereas no differences were observed in neurons from males (Krebs $0.1944 \pm 0.186$ vs. AA $0.1987 \pm 0.245$, $P = 0.128$).

Data from the preceding experiments suggest that S100B production regulates ongoing ENS activity but whether these effects are primarily the result of changes in glial excitability or extracellular functions of S100B on enteric neurons is unclear. Extracellular S100B has multiple effects on neurons that include regulating $Ca^{2+}$ influx (Barger & Eldik, 1992), modulating the frequency of $Ca^{2+}$ spikes (Bancroft et al., 2022) and regulating rhythmic bursting among neurons in CPGs (Morquette et al., 2015a). To test the extracellular effects of S100B on myenteric neurocircuits, we incubated CMMPs with pentamidine (10 μM), which blocks interactions between S100B and and $Ca^{2+}$ at the $Ca^{2+}$/p53 binding site and binding to RAGE (Charpentier et al., 2008; Hartman et al., 2013) or S100B antibodies (1:1000) that bind to and neutralize S100B in the extracellular space (Fig. 5A and B). Similar to experiments with arundic acid, neither pentamidine nor anti-S100B antibodies changed the relative number of active cells (glia: Krebs $68.10 \pm 4.89$ vs. PTM $60.17 \pm 7.01$, $P = 0.506$, and S100B ab $75.50 \pm 4.51$, $P = 0.4968$; neurons: $85.45 \pm 2.18$ vs. PTM $87.57 \pm 3.09$, $P = 0.884$, and S100B ab $85.87 \pm 3.83$, $P = 0.950$) (Fig. 5C and G), but both modified the strength of responses among enteric neurons and glia. Neuron response amplitudes decreased by 41% and glial responses by 16% following treatment with either pentamidine or S100B antibodies (glia: Krebs $0.1324 \pm 0.005$ vs. PTM $0.1148 \pm 0.007$, $P = 0.0056$, and S100B ab $0.1021 \pm 0.006$, $P < 0.0001$; neurons: Krebs $0.2765 \pm 0.017$ vs. PTM $0.1661 \pm 0.010$, and S100B ab $0.1671 \pm 0.013$; $P < 0.0001$) (Fig. 5D and H). Splitting the data by sex revealed that effects of pentamidine on glia were driven by effects in males where glial responses decreased by 30% in amplitude (Krebs $0.1086 \pm 0.005$ vs. PTM $0.0822 \pm 0.004$, $P = 0.0002$) (Fig. 5E). The effects of pentamidine on neurons were more complex

and increased response amplitudes among female neurons (62.5%), but decreasing those in males (–32%) (male: Krebs $0.1944 \pm 0.012$ vs. PTM $0.1362 \pm 0.008$, $P < 0.0001$; female: Krebs $0.2465 \pm 0.016$ vs. PTM $0.3923 \pm 0.038$, $P = 0.0001$) (Fig. 5I and J). S100B neutralization by antibodies had selective effects on the female ENS and decreased both glial and neuronal response amplitudes by half (glia: Krebs $0.1576 \pm 0.014$ vs. S100B ab $0.0749 \pm 0.009$; neurons: Krebs $0.2465 \pm 0.016$ vs. S100B ab $0.1297 \pm 0.011$, $P < 0.0001$) (Fig. 5F and J). Taken together, these data show that intra- and extracellular S100B pools control enteric neuron and glial activity as reflected by $Ca^{2+}$ responses. S100B release and extracellular interactions regulate spontaneous activity in the female ENS, whereas S100B release and specific $Ca^{2+}$-S100B interactions modulate ENS activity in males.

The effects observed with pentamidine could result from how this drug interferes with RAGE ligands, interactions between S100B and $Ca^{2+}$ or by directly blocking interactions between S100B and RAGE. To differentiate between these possibilities, we repeated CMC recordings and recordings of spontaneous ENS activity in the presence of the high-affinity RAGE antagonist FPS ZM1 (1 μM) or RAGE antibodies, respectively (Deane et al., 2012). By contrast with the previous manipulations of S100B, FPS ZM1 did not affect CMC behaviour (N = 5 mice; oral amplitude: buffer $100 \pm 25.21$ vs. FPS ZM1 $79.79 \pm 17.52$; $P = 0.1504$; aboral amplitude: buffer $100 \pm 46.69$ vs. FPS ZM1 $105.8 \pm 12.88$, $P = 0.816$; oral integral: buffer $100 \pm 26.35$ vs. FPS ZM1 $74.23 \pm 27.62$, $P = 0.3171$; aboral integral: buffer $100 \pm 51.39$ vs. FPS ZM1 $128.5 \pm 79.15$, $P = 0.636$; frequency: buffer $100 \pm 14.12$ vs. FPS ZM1 $127.4 \pm 43.56$, $P = 0.280$; propagation velocity: buffer $100 \pm 52.83$ vs. FPS ZM1 $114.2 \pm 34.12$, $P = 0.600$) (Fig. 6A–C). This observation was supported by immunolabeling experiments that showed an absence of RAGE expression in the myenteric plexus of the healthy colon (Fig. 6D). This supports the idea that pentamidine effects were due mainly to changes in S100B-$Ca^{2+}$ interactions. $Ca^{2+}$ imaging experiments also showed no effects on enteric neuron excitability following incubation with anti-RAGE antibodies (Krebs $0.258 \pm 0.27$ vs. RAGE ab $0.176 \pm 0.14$, $P = 0.0566$) (Fig. 6E and F). Therefore, pentamidine most probably exerts its effects on ENS activity in healthy myenteric plexus by $Ca^{2+}$ regulation in the extracellular space.

### Glial communication controls Ca²⁺ responses among enteric neurons and glia in enteric CPGs

The glial syncytium is fundamental for neuronal bursting and synchronization during rhythmic behaviours (Condamine et al., 2018). Signalling through glial networks in the brain and ENS is mediated, in part, through connexin-43 (Iacobas et al., 2006; McClain

et al., 2014; Stout et al., 2002). Therefore, we investigated whether disrupting glial signalling with the connexin-43 hemichannel mimetic peptide 43Gap26 would produce effects similar to S100B inhibition (Fig. 7*A* and *B*). Blocking connexin-43 hemichannels increased the number of neurons and glia that exhibited spontaneous activity (glia: Krebs $60.6 \pm 5.1$ *vs.* 43Gap26 $78.6 \pm 7.7$, $P = 0.059$; neurons: Krebs $88.7 \pm 3.1$ *vs.* 43Gap26

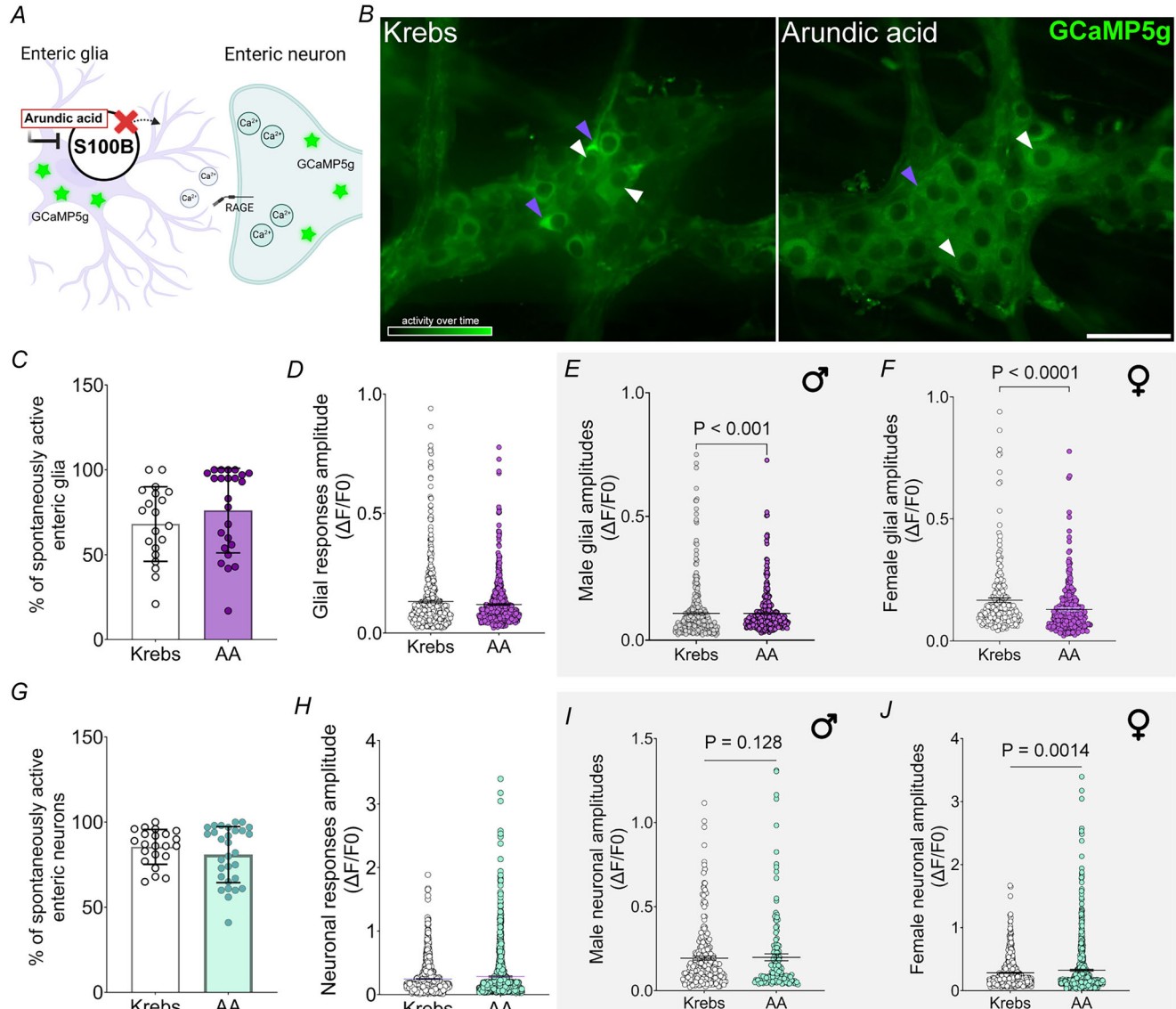

**Figure 4. S100B release controls spontaneous activity among myenteric neurons and glia in a sex-specific manner**

*A*, schematic representation showing mechanisms by which arundic acid (50 μM) inhibits S100B release in circular muscle-myenteric plexus whole-mount preparations. *B*, representative images of spontaneous activity over time in ganglia from *Wnt1*^Cre2GCaMP5g-tdTom^ mice under control conditions (Krebs buffer) and after arundic acid treatment. Purple arrows indicate enteric glia and white arrows indicate enteric neurons. Effects of arundic acid on the percentages of spontaneously active glia (*C*) and neurons (*G*) and response amplitudes (*D* and *H*) when data were pooled from both sexes. Stratifying responses by sex shows effects of arundic acid on male and female glial ($P < 0.001$, and $P < 0.0001$, respectively) (*E* and *F*) and neuronal response amplitudes ($P = 0.128$ and $0.0014$, respectively) (*I* and *J*). N = 10 mice; n = 720–1000 neurons and 600–650 glial cells. Data were analyzed using Student's *t* test, followed by the Mann–Whitney *post hoc* test. Scale bar = 50 μm. AA = arundic acid.

98.0 ± 1.2, *P* = 0.0029) (Fig. 7*C* and *G*); however, responses in these cells were significantly smaller than in control conditions (glia: Krebs 0.1467 ± 0.008 *vs.* 43Gap26 0.1338 ± 0.007, *P* = 0.001; neurons: 0.2502 ± 0.011 *vs.* 43Gap26 0.1922 ± 0.008, *P* = 0.0001) (Fig. 7*D* and *H*). Stratifying data by sex revealed that effect of 43Gap26 on neuron and glial response amplitudes were

driven by changes in samples from females mice (glia: Krebs 0.1800 ± 0.010 *vs.* 43Gap26 0.1006 ± 0.007, *P* < 0.0001; neurons: Krebs 0.2888 ± 0.015 *vs.* 43Gap26 3.199 ± 3.06, *P* < 0.0001) (Fig. 7*E*, *F*, *I* and *J*). This finding is consistent with the responses of female glia to S100B release inhibition but contrasts with the effects on enteric neurons. Connexin-43 hemichannels

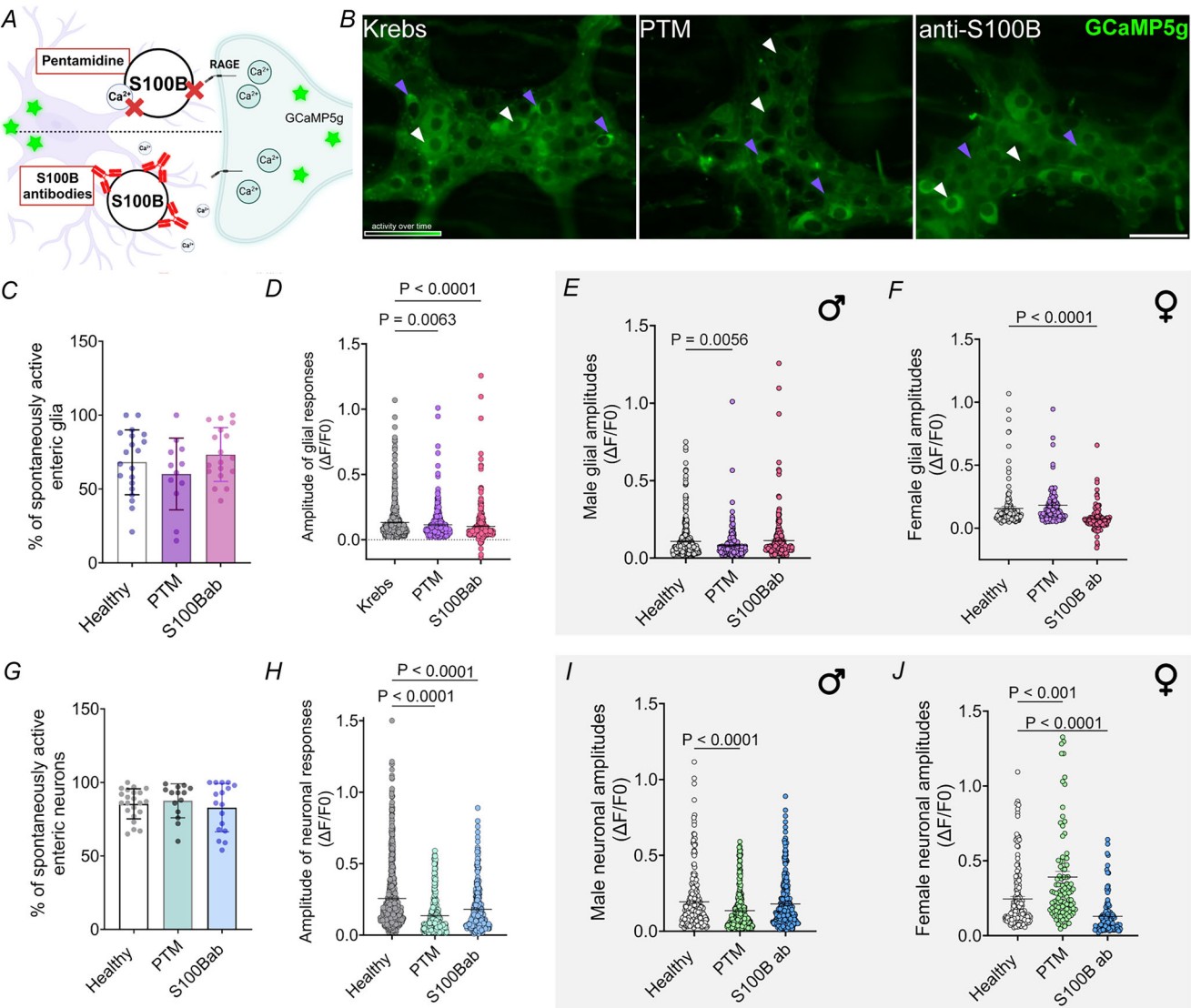

**Figure 5. Extracellular S100B–Ca$^{2+}$ interactions modulate glial and neuronal activity**
*A*, schematic illustrating two approaches to disrupt extracellular S100B signalling using pentamidine (PTM) that interferes with S100B–Ca$^{2+}$ interactions and subsequent binding to targets such as RAGE, and an anti-S100B-neutralizing antibody. *B*, representative images of activity over time in ganglia from *Wnt1*$^{Cre2GCaMP5g-tdTom}$ transgenic mice under control conditions (Krebs) and in ganglia following incubation with pentamidine or anti-S100B antibodies. Purple arrows indicate enteric glia and white arrows indicate enteric neurons. Effects of S100B-targeted interventions on the percentage of spontaneously active glia (*C*) and neurons (*G*) and their response amplitudes (*P* = 0.0063 and *P* < 0.0001; *P* < 0.0001 and *P* < 0.0001, respectively) (*D* and *H*). Sex stratified data showing effects of PTM and anti-S100B antibodies on male and female glial (*P* = 0.0056 and *P* < 0.0001, respectively) (*E* and *F*) and neuronal spontaneous response amplitudes (*P* < 0.0001; *P* < 0.001 and *P* < 0.0001, respectively) (*I* and *J*). *N* = 7 mice; *n* = 400–720 neurons and 400–600 glial cells. Data were analyzed using the Kruskal–Wallis test followed by Dunn's multiple comparisons test. Scale bar = 50 μm. PTM = pentamidine, S100B ab = S100B antibodies.

have been proposed as a pathway for S100B release (Condamine et al., 2018) and are also involved in how glia release multiple other transmitters in the ENS (Brown et al., 2016; Meunier et al., 2017) and so the effects of multiple pathways are probably altered by blocking connexin-43 hemichannels. Nevertheless, the data agree with the conclusion that glial signalling mediated by Cx43 hemichannels is required for physiological ENS signalling.

### S100B regulates spontaneous activity among ChAT(+) neurons in the ENS

Recordings of cellular activity in $Wnt1^{Cre2GCaMP5g\text{-}tdTom}$ mice give a broad assessment of the effects of S100B on all enteric neurons. However, the ENS is composed of multiple functional neuron subtypes, and it is reasonable

to expect that S100B has differential effects on these neurons if S100B is involved in regulating synaptic pathways. We therefore considered whether S100B exerts control over specific neuron subpopulations involved in CMC activity. We chose to focus on cholinergic neurons given their central role in regulating CMC occurrence and synchronization (Spencer et al., 2005; Spencer, Costa et al., 2021). Toward this end, we generated the $ChAT^{CreGCaMP5g\text{-}tdTom}$ mice by expressing GCaMP5g selectively in cholinergic neurons to study the effects of S100B on this subpopulation (Fig. 8A).

ChAT(+) neurons exhibited spontaneous activity, with some displaying a rhythmic and consistent firing pattern (Fig. 8B). Impairing S100B release with arundic acid increased the amplitude of spontaneous responses in ChAT(+) neurons by 47% (Krebs 0.3094 ± 0.010 vs. AA 0.4534 ± 0.016, $P < 0.0001$) (Fig. 8C and F) and

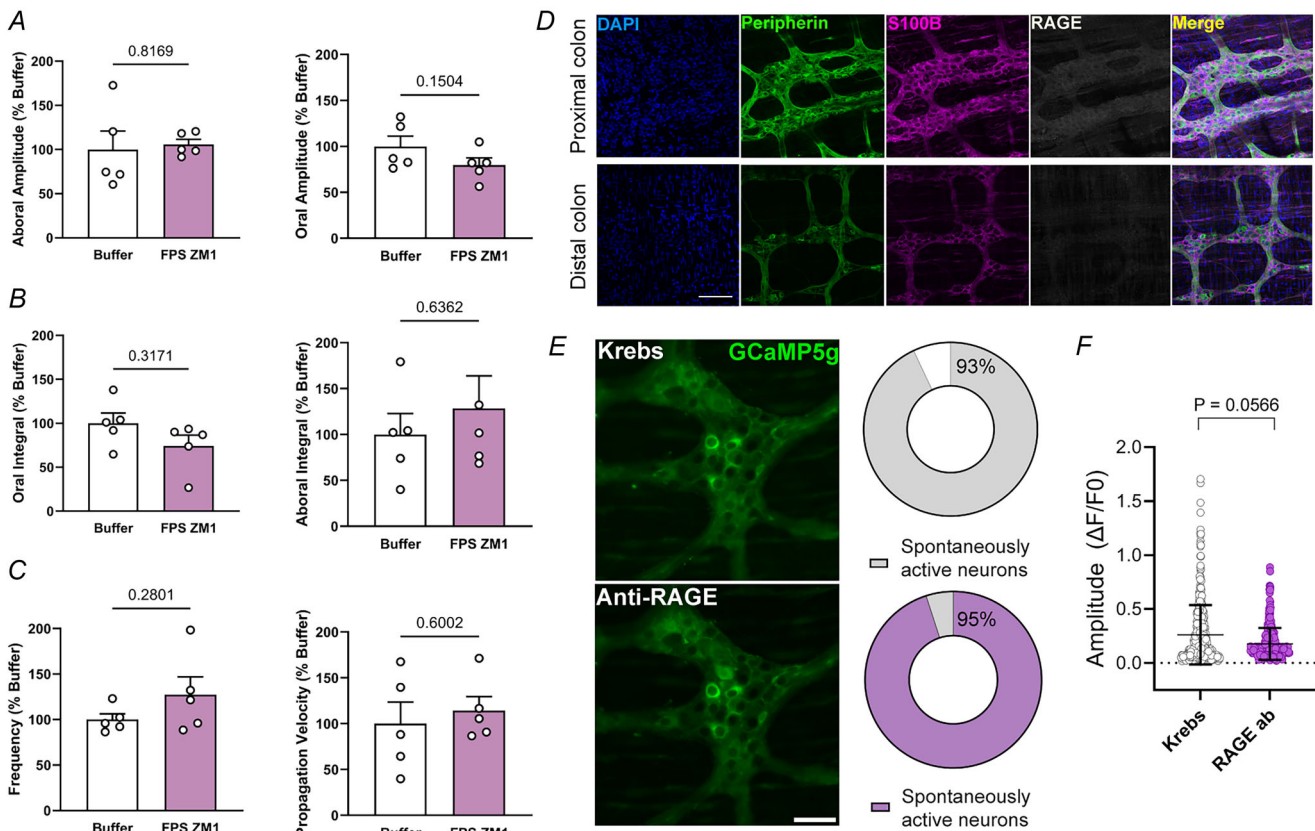

**Figure 6. Effects of S100B on colonic motor complexes are independent of RAGE signalling in the healthy colon**
Summary data showing that the RAGE antagonist FPS-ZM1 does not affect colonic motor complex amplitudes ($P = 0.8169$ and $0.1504$) (A), integral ($P = 0.3171$ and $0.6362$) (B), frequency ($P = 0.2801$) and propagation velocity ($P = 0.6002$) in colons from healthy mice (C). N = 4–5 mice for organ bath experiments. Data were analyzed using non-paired Student's t test. D, representative images of immunolabeling for DAPI (blue), peripherin (green), S100B (magenta) and RAGE (greyscale) in myenteric ganglia from the proximal and distal colons of mice showing that RAGE labelling is scarce in both regions during physiological conditions. Scale bar = 100 μm. E–F, effects of anti-RAGE-neutralizing antibodies on spontaneous neuronal activity in myenteric ganglia from $Wnt1^{Cre2GCaMP5g\text{-}tdTom}$ mice ($P = 0.0566$). N = 2 mice; n = ~300–400 neurons. Data were analyzed using Student's t test, followed by the Mann–Whitney post hoc test. Scale bar in (E) = 50 μm. RAGE ab = RAGE antibodies.

caused major changes to ChAT(+) response profiles (Fig. 8C). ChAT(+) neurons that normally exhibited rhythmic high- and low-frequency profiles lost this feature following exposure to arundic acid and despite exhibiting larger amplitude responses, these responses occurred at a lower frequency (−36%) (Krebs 0.0588 ± 0.002 *vs.* AA 0.0379 ± 0.001, $P < 0.0001$) (Fig. 8G). Importantly, effects of arundic acid were reversible and activity among ChAT(+) neurons was restored following a washout

period (AA 0.4534 ± 0.016 *vs.* post-AA 0.3542 ± 0.011, $P = 0.0003$) (Fig. 8D, F and G). Subsequent addition of recombinant S100B (50 μg mL$^{-1}$) did not produce major effects, despite a slight decrease in neuronal amplitudes (Krebs 0.3094 ± 0.010 *vs.* rS100B 0.3062 ± 0.015, $P = 0.30$) (Fig. 8E, F and G) and frequency (Krebs 0.0588 ± 0.002 *vs.* rS100B 0.04380 ± 004, $P = 0.0006$) (Fig. 8E, F and G) post-arundic acid. These data suggest that glial S100B constrains and refines activity among

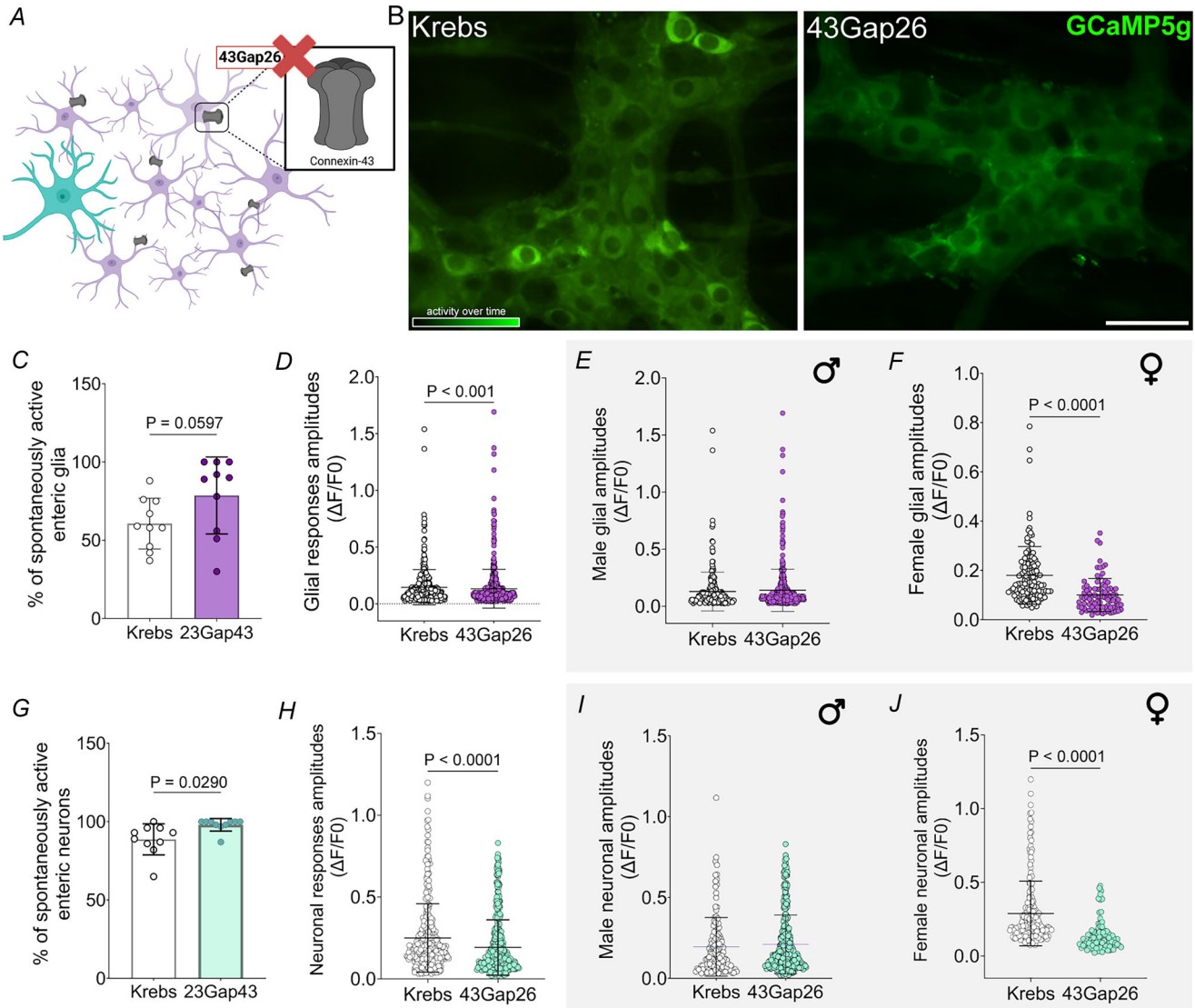

**Figure 7. Blocking glial hemichannels disrupts normal patterns of spontaneous activity among myenteric neurons and glia**

*A*, cartoon depicting how the connexin-43 hemichannel mimetic peptide 43Gap26 impairs glial network signalling. *B*, representative images of control (Krebs) and 43Gap26-treated (43Gap26) ganglia from *Wnt1*$^{Cre2GCaMP5g-tdTom}$ mice showing spontaneous activity over time. Effects of 43Gap26 on percentages of spontaneously active glia ($P = 0.0597$) (*C*) and neurons ($P = 0.0290$) (*G*) and their response amplitudes (pooled sexes, $P < 0.001$ and $P < 0.0001$, respectively) (*D* and *H*). Data stratified by sex showing effects of 43Gap26 on female glial ($P < 0.0001$) (*E* and *F*) and neuronal response amplitudes ($P < 0.0001$) (*I* and *J*). N = 4 mice; n = 400–800 neurons and n = 330–420 glial cells. Data were analyzed using Student's *t* test, followed by the Mann–Whitney *post hoc* test. Scale bar = 50 μm.

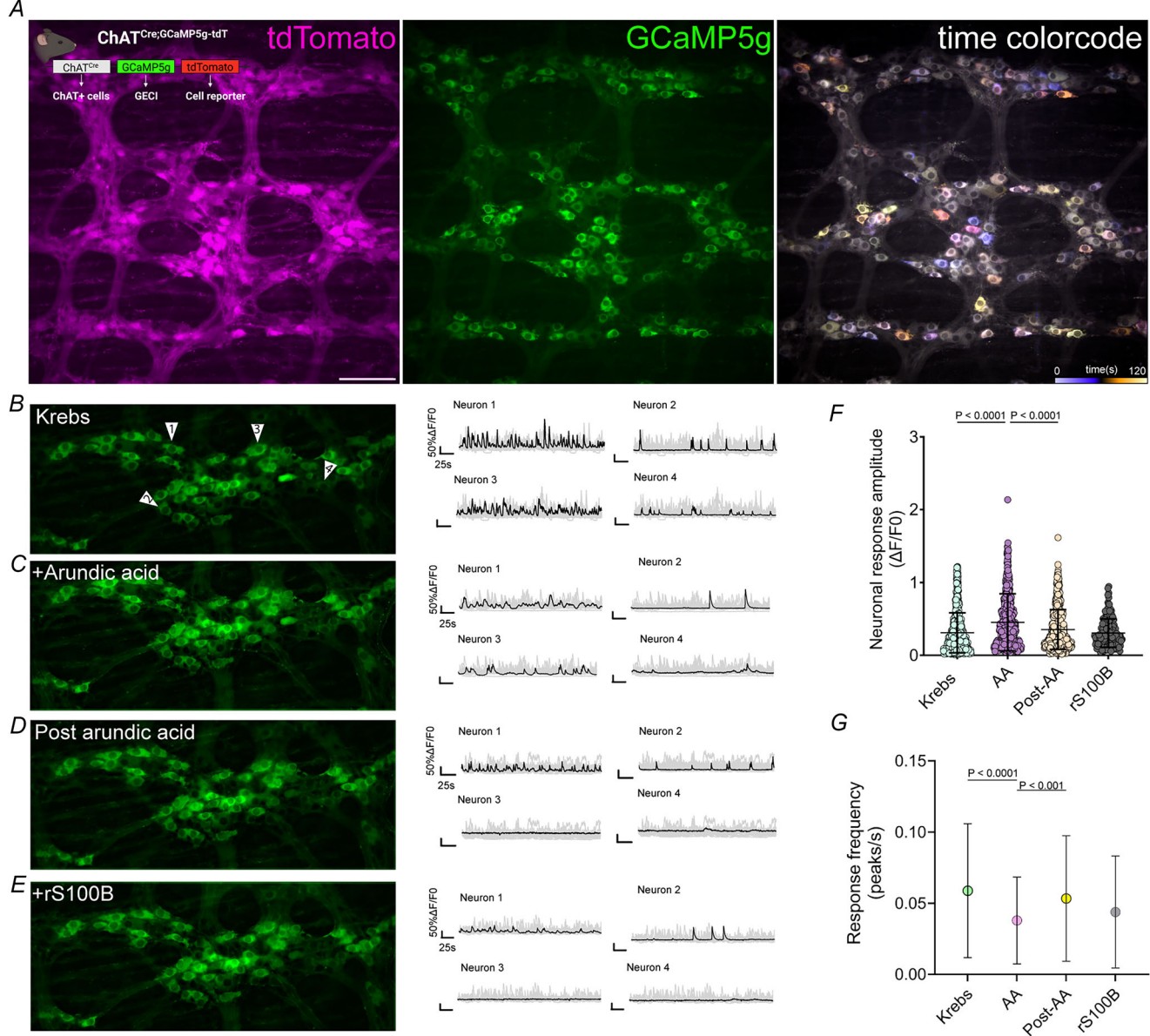

**Figure 8. S100B regulates spontaneous behaviours of cholinergic neurons in the colonic myenteric plexus**

*A*, representative images of tdTomato (magenta, left), GCaMP5g (green, middle), and spontaneous activity (temporally colour-coded image, right) in a myenteric ganglion from a *Chat*^CreGCaMP5g-tdTom mouse. Inset (left) shows a schematic of the *Chat*^CreGCaMP5g-tdTom transgenic line, which expresses the genetically encoded calcium indicator GCaMP5g specifically in ChAT(+) neurons.*B–E*, representative maximum intensity projections showing spontaneous neuronal activity over time under control conditions (Krebs) (*B*), following incubation with arundic acid (*C*), post-arundic acid washout (*D*) and addition of recombinant S100B (rS100B) (*E*). Representative traces of neuron activity are shown on the right. *F* and *G*, summary data showing quantification of effects on neuronal response amplitudes during arundic acid incubation (*F*) (*P* < 0.0001) and post-AA (*P* < 0.0001) and neuronal firing frequency during and post-AA (*G*) (*P* < 0.0001 and *P* < 0.001, respectively). *N* = 360–390 ChAT(+) neurons for Krebs, arundic acid and post-arundic acid; *N* = 100 neurons for rS100B. Data were analyzed using the Kruskal–Wallis test followed by Dunn's multiple comparisons test. Scale bar = 50 μm. AA = arundic acid, rS100B = recombinant S100B protein.

cholinergic neurons under physiological conditions, which is expected to have major effects on timing and co-ordination of rhythmic outputs (Berezhnov et al., 2021; Schnell et al., 2016; Shah et al., 2022; Yu et al., 2018).

## S100B release controls functional connectivity within enteric excitatory neurocircuits

Our data show that glial S100B controls fundamental aspects of neuron excitability, such as response amplitude,

frequency and patterns of activity. Functional interactions among neurons in CPGs are also crucial to ensure synchronization, modulation and resilience. Cholinergic neurons are the main excitatory drivers, phase co-ordinators, and regulators of synaptic transmission in CPGs (Lima et al., 2019; Perrins & Roberts, 1995; Ray et al., 2022; Spencer, Costa et al., 2021). To test how glial S100B regulates functional connectivity of ChAT(+) neurons in enteric CPG networks, we conducted a spatiotemporal analysis of pairwise correlations among cholinergic neurons in myenteric plexus.

Spontaneous activity among cholinergic neurons in the myenteric plexus is diverse under physiological conditions (Fig. 9A and B). Temporally colour-coded images of activity over time show variability of neighboring neurons with specific timing and intensity, but some

local correlations do exist (Fig. 9A and B). However, following treatment with arundic acid, neuronal pairs exhibited higher peak intensities and reduced diversity (Fig. 9A and B). Pairwise correlation analysis of $Ca^{2+}$ activity further revealed that S100B inhibition constrains ChAT(+) neurons into less diverse clusters (Fig. 9B). Representative dendrograms of ganglia from control and arundic acid-treated conditions demonstrated fewer clusters and greater distances between them, suggesting that S100B inhibition leads to a redundant neuronal correlation. Quantification confirmed that inhibiting S100B release significantly reduced the number of clusters (Krebs $6 \pm 0.5$ vs. AA $4 \pm 0.4$, $P = 0.007$) (Fig. 9C). This effect was apparent when viewing maps of activity over time (Fig. 9D), which clearly showed differing degrees of activity and higher intensities among some

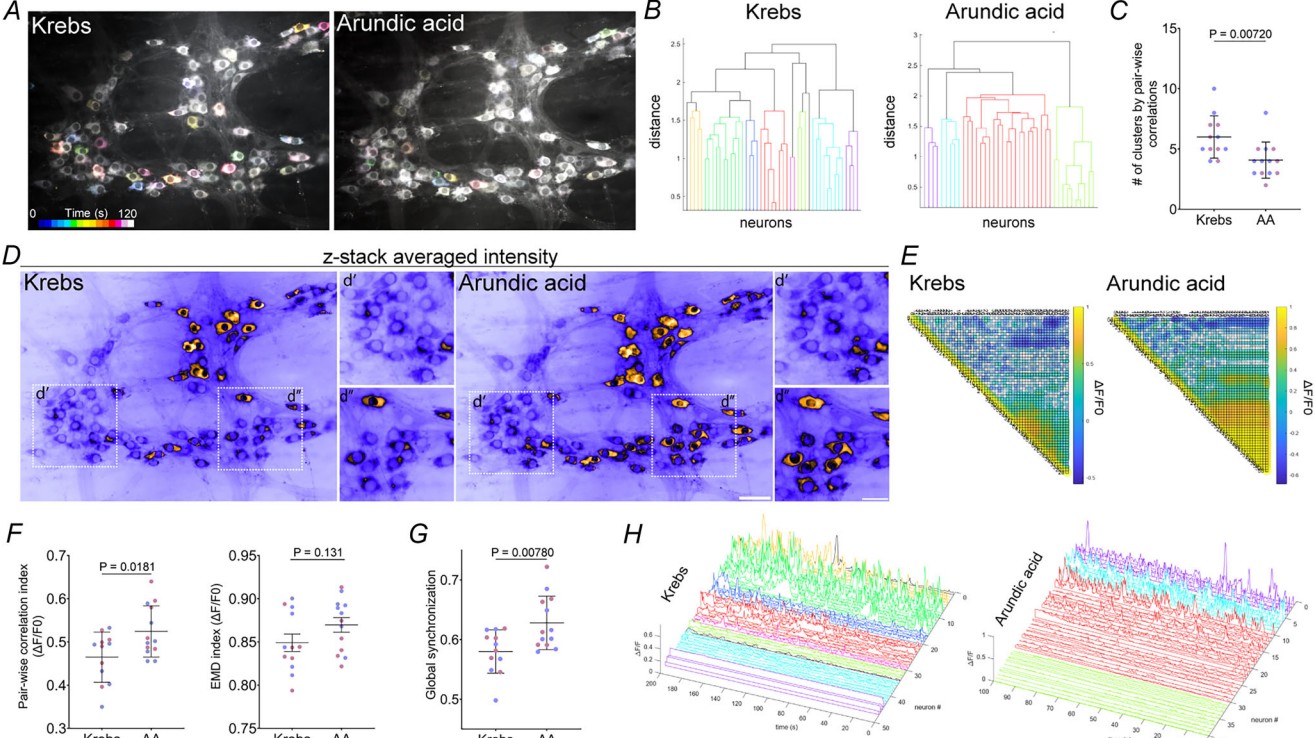

**Figure 9. Impairing S100B release modulates correlation and synchronization patterns among cholinergic neurons in the myenteric plexus**
*A*, temporally colour-coded images showing spontaneous activity among cholinergic neurons in a myenteric ganglion from a *Chat*$^{CreGCaMP5g-tdTom}$ mouse under control conditions (Krebs) and following incubation in arundic acid. Note that inhibiting S100B release reduces the diversity of spontaneous oscillations in ChAT(+) neurons over time. This effect is illustrated by dendrograms (*B*) that show clusters of ChAT(+) neurons grouped based on pairwise correlations. Summary data (*C*) reveal treating samples with arundic acid produces fewer distinct clusters among ChAT(+) neurons (*P* = 0.00720). *D*, Z-stacked average intensity images of activity over time show pairs of neurons becoming highly active and correlated following S100B inhibition. Zoomed regions (*Dd'* and *Dd''*) highlight similar response patterns between neuron pairs. *E*, representative heatmaps displaying the strength and extent of pairwise correlations among cholinergic neurons show that arundic acid increases correlated (green) and highly correlated (yellow) neuron pairs. *F* and *G*, summary data showing effects of arundic acid on normalized pairwise correlations (*P* = 0.0181) and Earth Mover's Distance (EMD) indices (*P* = 0.131) (***F***) and their combination (global synchronization) (*G*) (*P* = 0.00780). *H*, representative waterfall plots showing effects of arundic acid on temporal activity patterns of clustered cholinergic neurons. Ganglia: *N* = 12 or 13 from four animals; *n* cells = ∼500. Data were analyzed using an unpaired Student's *t* test. Scale bars = 50 μm (*A* and *D*); 25 μm (*d'* and *d''*). AA = arundic acid.

neuronal pairs (Fig. 9$Dd'$ and $Dd''$). Further analyses demonstrated pairwise cross-correlations increased between cholinergic neurons when S100B release is impaired (Krebs 0.464 ± 0.016 *vs.* AA 0.524 ± 0.016, $P = 0.0181$) (Fig. 9$E$ and $F$). Calculations of Earth Mover's Distance (EMD), a measure of frequency oscillation patterns, indicated negligible effects of S100B on specific-frequency patterns of neural activity (Krebs 0.849 ± 0.010 *vs.* AA 0.8697 ± 0.008, $P = 0.131$) (Fig. 9$F$); however, computations of the global synchronization index showed that inhibiting S100B release increased global synchronization among ChAT(+) neurons (Krebs 0.580 ± 0.010 *vs.* AA 0.6280 ± 0.012, $P = 0.0078$) (Fig. 9$G$). Representative waterfall traces from clusters evidence neuronal behaviours through time and the loss of diverse patterns (Fig. 9$H$). Overall, these data demonstrate that glial S100B is a mediator that maintains proper functional connectivity between cholinergic enteric neurons in gut neurocircuits.

### Extracellular S100B domains regulate myenteric functional connectivity

Glial regulation of the extracellular space and, notably, interactions with extracellular $Ca^{2+}$ ions are a key mechanism proposed for rhythmogenesis (Morquette et al., 2015$a$). Based on this, we hypothesized that extracellular actions of S100B are important for its modulatory effects on functional connectivity among cholinergic neurons. We tested this hypothesis by using anti-S100B antibodies to neutralize extracellular actions of S100B (Esposito et al., 2007; Mizuno et al., 2018; Morquette et al., 2015$a$).

Similar to effects observed when S100B release was inhibited with arundic acid, anti-S100B antibodies decreased the variability of correlations among ChAT(+) neuron responses and the numbers of clusters representing responses (Krebs 7.55 ± 4.2 *vs.* S100B ab 4 ± 0.81, $P = 0.0179$) (Fig. 10$A$–$C$). Again, connectivity maps in representative ganglia clearly indicated fields with pairs of neurons exhibiting increased correlation (Fig. 10$D$), which was also clear in data represented by correlation strength heatmaps (Fig. 10$E$). Normalized pairwise correlations increased following the application of anti-S100B antibodies (Krebs 0.4703 ± 0.046 *vs.* S100B ab 0.5248 ± 0.0517, $P = 0.0251$) (Fig. 10$F$); however, no differences were observed in the EMD index (Krebs 0.8776 ± 0.027 *vs.* S100B ab 0.8736 ± 0.044, $P = 0.819$). Similar to the effects of arundic acid, blocking extracellular S100B enhanced global synchronization of cholinergic neurons in enteric neurocircuits (Krebs 0.5928 ± 0.037 *vs.* S100B ab 0.6322 ± 0.040, $P = 0.0419$) (Fig. 10$G$). These findings suggest that both the release dynamics and extracellular actions of S100B are key glial mechanisms that shape neuronal functional connectivity in myenteric circuits. Specifically, S100B appears to exert an inhibitory effect that prevents excessive and overfitting coupling among ChAT(+) neurons, thereby promoting more diverse and dynamic connection patterns over time.

## Discussion

Fundamental patterns of gut motility are controlled by enteric neural networks that display characteristics of CPGs. Factors that modulate these circuits remain poorly understood. Here, we show that S100B, a protein often used as a marker for enteric glia but with unknown physiological significance, plays a major role in modulating activity within enteric neural networks that drive CMC motor patterns. Our data uncover a surprising dependence of CMCs on S100B and show that the mechanisms underlying this effect involve extracellular actions that modulate neuronal and glial activity. Additionally, we show that S100B functions as an enteric gliotransmitter that shapes functional connectivity among cholinergic neurons that are critical for CMC function in myenteric circuits. S100B suppresses redundant firing between pairs of cholinergic neurons and promotes a more controlled pattern of excitatory signalling. Together, these observations uncover glial S100B as a key neuromodulator in gut motor neurocircuits that contributes to an inhibitory tone permissive for CMC behaviours. Understanding these mechanisms of glial neuromodulation could be important to understand common gut diseases.

S100B is a $Ca^{2+}$-binding protein that is produced almost entirely by enteric glia in the intestine. Although S100B has gained attention for potential roles in gut pathophysiology, how it might regulate the normal physiology of enteric neurocircuits has remained unclear. S100B is synthesized and released by glial cells in the nervous system and acts to regulate intra- and extracellular $Ca^{2+}$ concentrations in both neurons and glia (Barger & Eldik, 1992; Morquette et al., 2015). These actions have a major influence over neural network excitability given the central roles played by $Ca^{2+}$ in cell signalling. Intracellular S100B also has important effects on glial protein phosphorylation and cytoskeletal dynamics (Garbuglia et al., 1998; Ivanenkov et al., 1995; Kursula et al., 1999; Sorci et al., 2000). S100B disrupts tubulin and intermediate filaments (Davey et al., 2001; Donato, 1988; Sorci et al., 2000) and inhibits glial fibrillary acidic protein phosphorylation (Frizzo et al., 2004). Cytoskeletal dynamics and $Ca^{2+}$ activity are intrinsically linked (Cotrina et al., 1998), which places S100B in a critical position to regulate glial intracellular milieu and $Ca^{2+}$ responses. Our data show that enteric glia exhibit spontaneous $Ca^{2+}$ responses that are modulated both

by the release and extracellular activity of S100B. Interestingly, extracellular blockade of S100B produced the same effect on glial activity as that observed when glial S100B production and release were inhibited by arundic acid. These findings suggest that S100B may regulate glial excitability through multiple mechanisms that include intracellular and extracellular targets. The resulting effects of S100B on glial $Ca^{2+}$ signalling mechanisms would be expected to impact glial behaviours and gliotransmission mechanisms by which enteric glia modulate neuronal function (Agulhon et al., 2012; Goenaga et al., 2023). Therefore, effects of S100B on enteric neuronal activity could conceivably occur subsequent to effects on glial excitability and gliotransmission or direct effects

on neurotransmission that involve extracellular $Ca^{2+}$ buffering. Therefore, effects of S100B on enteric neuronal activity could conceivably occur subsequent to effects on glial excitability and gliotransmission or direct effects on neurotransmission that involve extracellular $Ca^{2+}$ buffering. Direct effects of arundic acid on neurons have not been reported in studies of neurons in cell culture or ENS development (Asano et al., 2005; Hao et al., 2017; Tateishi et al., 2002); however, we cannot completely rule out the possibility that some of the effects we observed were due to direct actions on neurons. Given the similar effects of multiple drugs and antibody sequestering strategies on CMC function and ENS cell excitability, we consider that our data are most consistent with glial

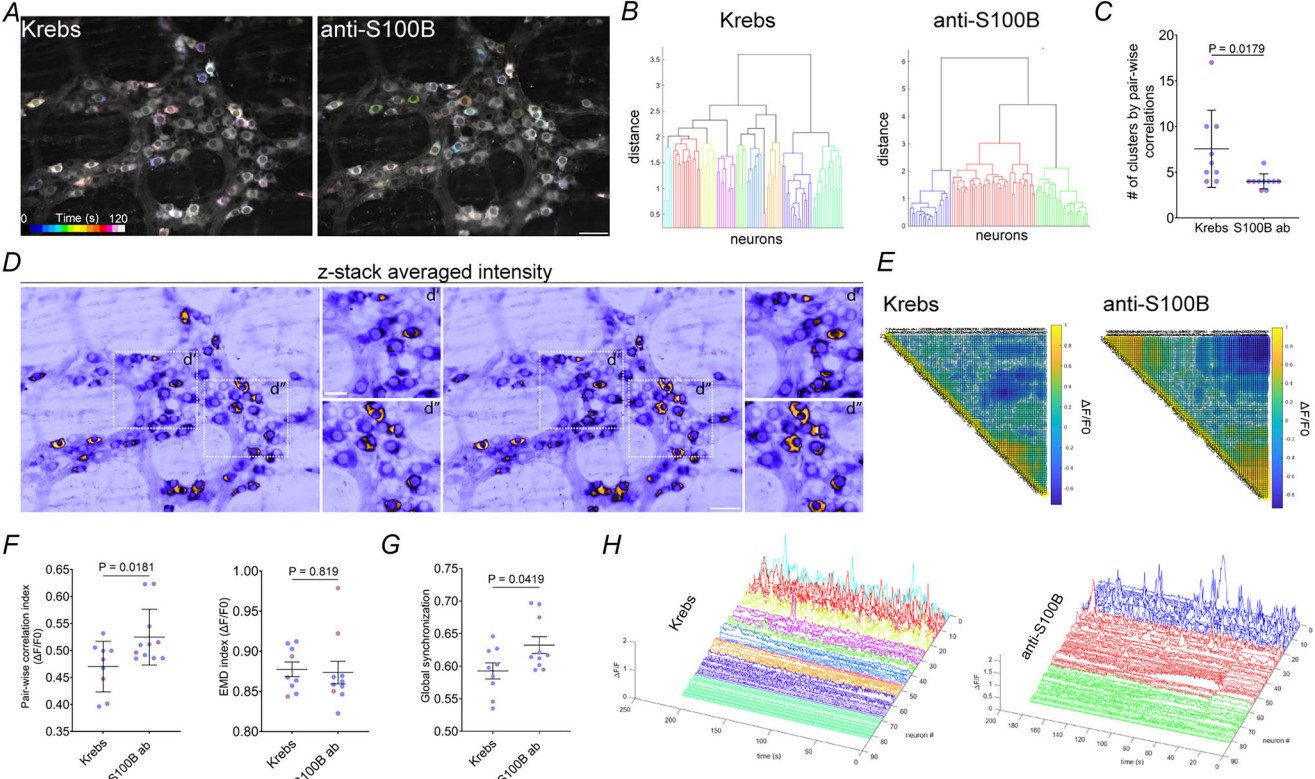

**Figure 10. Neutralization of extracellular S100B enhances pairwise correlations among cholinergic myenteric neurons**
*A*, temporally colour-coded images of spontaneous activity among ChAT(+) neurons in a myenteric ganglion from a *Chat*$^{CreGCaMP5g-tdTom}$ mouse under control conditions (Krebs) or following exposure to anti-S100B-neutralizing antibodies. Dendrograms showing clustering based on pairwise correlation coefficients (*B*) and summary quantification data (*C*) show that blocking extracellular S100B reduces the diversity of spontaneous ChAT(+) neuron oscillations (*P* = 0.0179). *D*, representative images of cholinergic neuron activity over time (z-stacked average intensity) show that neutralizing extracellular S100B increases strongly correlated and coactive neuron pairs. Zoomed regions (*Dd′* and *Dd″*) further highlight increased activity and pairwise correlations. Heatmaps (*E*) show that neutralizing extracellular S100B increases the strength and extent of correlations, with enhanced connectivity among neuron pairs. *F* and *G*, summary data showing that blocking extracellular S100B increases indices of normalized pairwise correlations (*P* = 0.0251) and Earth Mover's Distance (EMD) indices (*P* = 0.819) (*F*) and the global synchronization index (*P* = 0.0419) among cholinergic neurons (*G*). *H*, representative traces showing activity among clustered ChAT (+) neurons in control and anti-S100B antibody treated samples. Ganglia: *N* = 8–10 from three animals; *n* cells = ∼350. Data were analyzed using an unpaired Student's *t* test. Scale bars = 50 μm (*A* and *D*); 25 μm (*d′* and *d″*). S100B ab = S100B antibodies.

regulation, and this would agree with prior data showing that S100B affects neurotransmission in the brain by regulating $[Ca^{2+}]_e$ (Morquette et al., 2015*a*).

The potential for S100B to modulate CPG activity through $Ca^{2+}$-buffering mechanisms was elegantly shown by Morquette et al. (2015*a*) who found that blocking S100B prevented neuronal bursting in the mastication centre, at the same time as applying exogenous S100B promoted rhythmogenesis by regulating $[Ca^{2+}]_e$. These, and other consistent observations have led to the concept that extracellular S100B regulates neuronal bursting patterns and contributes to rhythmicity in CPGs (Morquette et al., 2015*a*). Our data suggest a similar role for S100B in the ENS. Central to this argument are the observed effects of inhibiting S100B release on ChAT(+) myenteric neurons, which included decreased frequency of $Ca^{2+}$ responses. Cholinergic myenteric neurons drive the contractile phase of CMCs and regulating their activity is essential for generating periodic and cyclic colonic behaviours. Our data show that S100B constrains activity among cholinergic neurons and prevents neurons within myenteric circuits from becoming aberrantly correlated. Similar abnormal hyperactivity of enteric neurons disrupts gut motility following inflammation, which could suggest a role for S100B in this process (Linden et al., 2003; Mawe, 2015). These effects resemble the anti-epileptogenic function of S100B in the brain, where mice lacking S100B exhibit severe seizures with brain regions becoming hypercorrelated (Dyck et al., 2002; Jiruska et al., 2013). On the other hand, over-expression of S100B modifies synaptic plasticity in the hippocampus and leads to dysfunctional spatial learning (Gerlai et al., 1995). The mechanism by which S100B constrains cholinergic neural networks does not appear to involve RAGE receptor-dependent pathways under physiological conditions because blocking RAGE did not affect CMCs or neuronal activity. Rather, these effects are more probably associated with extracellular $Ca^{2+}$ regulation.

Disrupting glial communication by blocking Cx43 hemichannels produced effects that were at least partially consistent with those observed when S100B was blocked by arundic acid, which could suggest that S100B exerts its effects through glial signalling mechanisms involving Cx43 hemichannels. However, 43Gap26 produced stronger effects on neurons than arundic acid. One explanation for this difference is that specific effects of S100B inhibition may have been masked among Wnt1(+) cells, which include both excitatory and inhibitory neurons in these experiments. It is also probable that blocking Cx43 hemichannels produced more robust effects by impairing multiple modes of gliotransmission. Indeed, the release of several gliotransmitters is attributed to hemichannel-dependent mechanisms, particularly those mediated by Cx43 (Brown et al., 2016;

Grubisic et al., 2020; Morales-Soto et al., 2023; Rafael et al., 2024). Such Cx43-dependent gliotransmission mechanisms are important for how astrocytes release multiple gliotransmitters, which are thought to play complementary roles in CPGs through metamodulation (Broadhead & Miles, 2021). For example, rhythmic ATP released by glia, followed by conversion to ADP and then adenosine, controls cyclic and rhythmic responses in the locomotor system in a neurotransmission-dependent manner (Broadhead & Miles, 2020). In the respiratory CPGs, ATP itself creates excitatory rhythmic patterns by increasing neuronal spike frequency, while its conversion to adenosine inhibits rhythmicity by decreasing neuronal frequency (Gourine et al., 2010). Glutamate-driven phasic $Ca^{2+}$ responses between neurons and astrocytes and regulation over GABA in circadian sleep–wake cycles also highlight astrocytic-neuronal communication driving periodic responses (Brancaccio et al., 2017; Patton et al., 2023). Therefore, gliotransmission mechanisms that involve Cx43 contribute to multiple aspects of CPG function by modulating synaptic transmission, neuronal activity and, ultimately, network synchrony (Chever et al., 2016; Dyck et al., 2002; Gerlai et al., 1995; Jiruska et al., 2013; Pannasch et al., 2012).

Functional connectivity in neural networks provides the foundation for neural programs that underly complex and rhythmic functions. Several processes, such as synchronization of neuronal discharges, neuronal correlation and intrinsic heterogeneity, shape the coding properties of these networks (Padmanabhan & Urban, 2010; Gansel, 2022). In the ENS, increased activity of cholinergic neurons occurs during CMC generation (Debnath et al., 2025; Gould et al., 2019) (Debnath et al., 2025; Gould et al., 2019) and pulsatile acetylcholine release is thought to co-ordinate rapid oscillations during CMCs (Barth et al., 2023). Our data show that cholinergic neurons exhibit $Ca^{2+}$ responses that are diverse in their kinetics and that activity amongst these neurons is spatially and temporally segregated in myenteric neurocircuits. This is consistent with prior observations that classified non-, sporadic and constant-like activity patterns among cholinergic neurons during spontaneous activity (Debnath et al., 2025). Interestingly, intrinsic biophysical diversity shapes how neurons encode signals by reducing pairwise correlations and enhancing the informational capacity of the neuronal population (Padmanabhan & Urban, 2010). ChAT(+) myenteric neurons comprise several distinct functional subtypes of enteric neurons and the differing patterns of spontaneous activity observed in motoneurons and interneurons reflect varying levels of control within this heterogeneous network (Barth et al., 2023). Our data show that glial S100B plays an important role in regulating how information travels in cholinergic enteric neural networks. Enteric glia interact bidirectionally with

cholinergic neurons, in part through the expression of M3 muscarinic receptors, for which the activation elicits $Ca^{2+}$ responses in glial cells (Delvalle, Fried et al., 2018). Cholinergic enteric neuron-to-glia communication modulates several aspects of CMC function through mechanisms that involve refinement of motor circuits pathways (Ahmadzai et al., 2021a). Interestingly, blocking glial cholinergic activation produces a loss of specificity in motility pathways, which is similar to effects on ChAT(+) neurons observed here following S100B inhibition. This raises the possibility that enteric glia sense activity in cholinergic neurocircuits and secrete S100B as a mechanism to constrain and refine activity within these circuits. We should then consider that CMC dysfunction may involve altered spontaneous activity in ENS cells, hypercorrelated signalling inside the cholinergic network, and modified cholinergic signalling on glia. Neuronal hypercorrelation is a hallmark of dysfunctional neural networks (Jiruska et al., 2013; Raghavan et al., 2024) and hyperexcitability underlies post-inflammatory colon dysfunction (Mawe et al., 2009). Therefore, a further understanding of how glial S100B regulates neuron-to-neuron interactions and intrinsic inputs that culminate in the rhythmic properties of the colon will be important for a better understanding of gut health and disease. This may include gliotransmission effects into other subpopulations that are fundamental to CMC, such as nitrergic neurons. Specialized interactions between enteric glia and nitrergic neurons have been proposed for tonic inhibition and CMC generation periods (Smith & Koh, 2017) and may play a key role in pathological conditions where glial purinergic signalling contributes to nitrergic neuron death (Brown et al., 2016). In the same vein, glial signalling may influence interstitial cells of Cajal and the SIP (i.e. Smooth muscle cells, Interstitial cells of Cajal and Platelet-derived growth factor receptor alpha cells) syncytium by their purinergic signalling (Gulbransen & Sharkey, 2009a; Hendler et al., 2024; Montalant et al., 2021). Indirect effects on the smooth muscle environment may also occur through S100B glial modulation of enteric neurons, notably nitrergic subpopulations. In the CNS, S100B promotes neuronal death by astrocytic release of nitric oxide (Hu et al., 1997). Similarly, S100B–RAGE axis activation induces nitric oxide production (Villarreal et al., 2014) which may alter the balance of enteric neurotransmission and intercellular interactions, ultimately affecting motility outputs.

S100B is widely used as a glial marker in the gut but its role in gut physiology has remained unknown. The evidence we present here shows that S100B-mediated gliotransmission regulates ENS excitability and enteric CPG neural networks that control fundamental patterns of gut motility. S100B modulates both neuronal and glial excitability in the myenteric plexus, the strength of responses among neurons in these circuits, and

functional connectivity among cholinergic neurons. Inhibition of S100B disrupts regulatory control over ChAT(+) neuronal networks, resulting in increased signalling redundancy and heightened correlation among neurons. CMC recordings show that this ultimately leads to colon dysfunction. These findings uncover a novel glial mechanism that governs not only neuronal behaviours, but also connectivity within the ENS. Given that S100B is implicated in pathological conditions such as inflammatory bowel disease, post-infectious irritable bowel syndrome, and other neuroinflammatory disorders, it emerges as a promising target for disease intervention.

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

# Additional information

### Data availability statement

Main resources generated from this study, including mouse lines, analytic methods, study materials and results, will be made available upon request.

### Competing interests

The authors declare that they have no competing interests.

### Author contributions

B.T. and B.G. were responsible for conceptualization, methodology, writing the original draft and reviewing and editing. B.T. was responsible for investigations, validation, data curation, formal analysis, visualization and project administration. J.J. and R.L. were responsible for data curation and investigations. J.L.M. was responsible for investigationd and reviewing and editing. L.S. was responsible for visualization and reviewing and editing. B.D.G. was responsible for funding acquisition, resources and supervision. All authors have approved the final version of the manuscript submitted for publication. They agree to be accountable for all aspects of the work in ensuring that questions related to the accuracy or integrity of any part of the work are appropriately investigated and resolved. Furthermore, all persons designated authors qualify for authorship, and all those who qualify for authorship are listed.

### Funding

This work was supported by grants R01DK103723 and R01DK120862 to BDG from the National Institute of Diabetes and Digestive and Kidney Diseases (NIDDK) of the National Institutes of Health (NIH). The content is solely the responsibility of the authors and does not necessarily represent the official views of the NIH.

### Acknowledgements

We thank Professor Roberto De Giorgio (Medical Sciences, University of Ferrara) for generously providing the healthy human tissue samples used in this study.

### Keywords

central pattern generators, colonic motor complex, enteric glia, enteric nervous system, S100 calcium-binding protein B

## Supporting information

Additional supporting information can be found online in the Supporting Information section at the end of the HTML view of the article. Supporting information files available:

**Peer Review History**

