## [Peer Review History · The Journal of Physiology]

Enteric glial S100B controls rhythmic colonic functions by regulating excitability and specificity in gut motor neurocircuits

Beatriz Thomasi, Rafaella Lavalle, Jonathon L. McClain, Julia R. Jamka, Luisa Seguella, and Brian David Gulbransen
DOI: 10.1113/JP289410

Corresponding author(s): Beatriz Thomasi (thomasi7@msu.edu)

Review Timeline:

Submission Date:	04-Jun-2025
Editorial Decision:	08-Jul-2025
Revision Received:	29-Jul-2025
Accepted:	13-Aug-2025

Senior Editor: Kim Barrett

Reviewing Editor: Bernard Drumm

Transaction Report:

Dear Dr Thomasi,

Re: JP-RP-2025-289410 "Enteric glial S100B controls rhythmic colonic functions by regulating excitability and specificity in gut motor neurocircuits" by Beatriz Thomasi, Rafaella Lavalle, Jonathon L. McClain, Julia R. Jamka, Luisa Seguella, and Brian David Gulbransen

Thank you for submitting your manuscript to The Journal of Physiology. It has been assessed by a Reviewing Editor and by 2 expert referees and we are pleased to tell you that it is acceptable for publication following satisfactory revision.

REVISION CHECKLIST:

We look forward to receiving your revised submission.

Yours sincerely,

Kim Barrett
Senior Editor
The Journal of Physiology

REQUIRED ITEMS

- Include a Key Points list in the article itself, before the Abstract.
- Author photo and profile. First or joint first authors are asked to provide a short biography (no more than 100 words for one author or 150 words in total for joint first authors) and a portrait photograph. These should be uploaded and clearly labelled together in a Word document with the revised version of the manuscript. See Information for Authors for further details.
- You must start the Methods section with a paragraph headed Ethical approval (https://jp.msubmit.net/cgi-bin/main.plex?form_type=display_requirements#methods).

Research must comply with The Journal's policies regarding animal experiments (<https://physoc.onlinelibrary.wiley.com/hub/animal-experiments>) and adherence to these policies must be stated in the manuscript.

Authors should confirm in their Methods section that their experiments were carried out according to the guidelines laid down by their institution's animal welfare committee, including an ethics approval reference number. The Methods section must contain a statement about access to food, water and housing, details of the anaesthetic regime: anaesthetic used, dose and route of administration, and method of killing the experimental animals.

- Your manuscript must include a complete Additional Information section, including competing interests; funding; author contributions and acknowledgements.

- Please upload separate high-quality figure files via the submission form.

- Papers must comply with the Statistics Policy: https://jp.msubmit.net/cgi-bin/main.plex?form_type=display_requirements#statistics.

In summary:

- If $n \leq 30$, all data points must be plotted in the figure in a way that reveals their range and distribution. A bar graph with data points overlaid, a box and whisker plot or a violin plot (preferably with data points included) are acceptable formats.
- If $n > 30$, then the entire raw dataset must be made available either as supporting information, or hosted on a not-for-profit repository, e.g. FigShare, with access details provided in the manuscript.
- 'n' clearly defined (e.g. x cells from y slices in z animals) in the Methods. Authors should be mindful of pseudoreplication.
- All relevant 'n' values must be clearly stated in the main text, figures and tables.

- The most appropriate summary statistic (e.g. mean or median and standard deviation) must be used. Standard Error of the Mean (SEM) alone is not permitted.

- Exact p values must be stated. Authors must not use 'greater than' or 'less than'. Exact p values must be stated to three significant figures even when 'no statistical significance' is claimed.

- A Data Availability Statement is required for all papers reporting original data. This must be in the Additional Information section of the manuscript itself. It must have the paragraph heading 'Data Availability Statement'. All data supporting the results in the paper must be either: in the paper itself; uploaded as Supporting Information for Online Publication; or archived in an appropriate public repository. The statement needs to describe the availability or the absence of shared data. Authors must include in their statement: a link to the repository they have used, or a statement that it is available as Supporting Information; reference the data in the appropriate sections(s) of their manuscript; and cite the data they have shared in the References section. Whenever possible, the scripts and other artefacts used to generate the analyses presented in the paper should also be publicly archived. If sharing data compromises ethical standards or legal requirements then authors are not expected to share it, but must note this in their statement. For more information, see our Statistics Policy.

- Please include an Abstract Figure file, as well as the Figure Legend text within the main article file. The Abstract Figure is a piece of artwork designed to give readers an immediate understanding of the research and should summarise the main conclusions. If possible, the image should be easily 'readable' from left to right or top to bottom. It should show the physiological relevance of the manuscript so readers can assess the importance and content of its findings. Abstract Figures should not merely recapitulate other figures in the manuscript. Please try to keep the diagram as simple as possible and without superfluous information that may distract from the main conclusion(s). Abstract Figures must be provided by authors no later than the revised manuscript stage and should be uploaded as a separate file during online submission labelled as File Type 'Abstract Figure'. Please also ensure that you include the figure legend in the main article file. All Abstract Figures should be created using BioRender. Authors should use The Journal's premium BioRender account to export high-resolution images. Details on how to use and access the premium account are included as part of this email.

- The corresponding author must provide an institutional email address (not a personal address) for their author account. We encourage ALL co-authors to also provide institutional email addresses. If this cannot be provided (as corresponding author), then a stamped letter must be provided from the institution which confirms their role and employment there (please upload this with the revised submission).

EDITOR COMMENTS

Reviewing Editor:

Thank you for submitting your work to the Journal.

The two referees that have reviewed the paper both agree in general that the study is well performed, experimental approach is sophisticated, and data analysis and separation of data based on sex are both highly commendable features of the paper.

An important concern is that reviewer 1 points to an earlier study in the Journal (<https://pubmed.ncbi.nlm.nih.gov/22063626/>) which previously demonstrated glial cells interact with the ENS during colonic motor complexes (CMCs), and makes some of the same observations as the current study. The authors should cite this previous paper noted by reviewer 1 (<https://pubmed.ncbi.nlm.nih.gov/22063626/>) and explicitly provide evidence on how the current study expands on or differentiates itself from this previous work.

In addition, in agreement with reviewer 2, a discussion on how S100B signalling from glia might interact with other cell types would be most welcome. In the submitted paper, a clear emphasis is put on s100B effects on cholinergic neurons (for reasons the authors explain). However, activation of nNOS+ neurons are known to influence CMCs, which still occur in the presence of atropine (<https://pubmed.ncbi.nlm.nih.gov/35446689/>) through signalling via ICC. While any such discussions on

this would be speculative, a brief discussion would be beneficial.

Please also take careful note of the Journals statistics policy in any resubmission, as some summary graphs show exact p values for statistical significance and some do not. Refer here: https://jpp.msubmit.net/cgi-bin/main.plex?form_type=display_requirements#statistics

Also note that the exact anaesthetic protocol used for animal sacrifice must be stated in the Methods.

REFeree COMMENTS

Referee #1:

This study investigates role of glia cells in the generation of a pattern of motor activity that occurs in the large intestine, called the colonic motor complex (CMC). The authors use a transgenic mouse line which they have published with before (and validated) (Wnt1Cre2GCaMP5g-tdTom) which seems to be a good model to study enteric neuron and glial activity. The mouse line is established in the laboratory of the authors and segments of colon are removed from these transgenic animals and studied ex vivo. The authors also used a transgenic mouse that expressed the calcium indicator in excitatory neurons that are cholinergic in the enteric nervous system (ChatCreGCaMP5g-tdTom).

The mechanisms that generate CMCs is not fully understood but have been shown to be dependent upon the enteric nerves. The authors extend this understanding, and the basic premise of this study is that S100B is a likely candidate by which enteric glia modulate intrinsic neurons underlying CMC contraction generation along the colon.

S100B is a calcium binding protein primarily found in astrocytes, a type of glial cell that is found in the central nervous system. It is believed that S100B is localised in the cytoplasm and nucleus of a wide range of cells and is involved in cycle progression and differentiation. It has also been shown to be involved in neuronal and endothelial dependent cerebral arteriolar dilation.

The study shows that CMC activity was almost blocked when spontaneous S100B release was impaired by adding arundic acid. This drug is believed to be specific for S100B. The authors show that S100B can be released and detected from segments of myenteric plexus with muscle attached.

Figure 9 shows synchronisation patterns of firing of cholinergic neurons is altered after blocking S100B. This is an interesting observation, that supports other studies on the same preparation. The authors conclude that their observations show that CMCs are controlled by an enteric central pattern generator, that requires S100B through mechanisms that are likely to involve the extracellular Ca²⁺-binding actions of S100B, although how S100B actually works in the colon is not resolved in this study.

CONCERNS with this study:

1. My major concern with this study is that enteric glia have already been demonstrated to be involved in the motor behaviour of the colon of mice, called CMCs, see 10.1113/jphysiol.2011.219519. Hence, the novelty of the study is somewhat in doubt.

2. How do the authors know that arundic acid doesn't act on myenteric neurons?

3. The mechanism of action of pentamidine is not well understood. Without a clear understanding of how the drug works doesn't provide strong support that the drug is specific of enteric glia.

4. Figure 1 of reference <https://pmc.ncbi.nlm.nih.gov/articles/PMC5322270/> it has been shown that S100B expression colocalises with enteric neural crest cells that are SOX10 positive. Hence, the idea that S100B is exclusive to enteric glia and not expressed in the ENS is not unequivocal <https://doi.org/10.3389/fncel.2017.00042> S100B is expressed in some neurons of rat see (10.1016/0306-4522(94)00615-c), and in mice SB100 expression has been shown in astrocytes and neurons in specific parts of the brain see: <https://doi.org/10.1523/JNEUROSCI.12-11-04337.1992> The authors need to validate that S100B is not expressed in enteric neurons of mouse colon, using more than antibodies?

5. Line 113 states: based on the current data, this statement should be softened: "Enteric glia are unique neuroglia of the ENS and the only cellular source of S100B within gut motor neurocircuits." Again, what is the evidence no other cells express S100B in these preparations of gut?

6. A major reference by Broadhead M (2012) Journal of Physiology is not cited, that studied role of glia in CMCs. This should be quoted.

7. Figure legend 9 is slightly confusing. The title states "Impairing S100B release disrupts correlation and synchronization patterns among cholinergic neurons in the myenteric plexus." But figure 9B legend states that in "arundic acid increases correlated (green) and highly correlated (yellow) neuron pairs." Figure 9E shows more yellow which to me means increased correlation of neuronal pairs. Please clarify what this is showing.

Line 374: rather than " caused a major failure of CMC activity", may I suggest "caused a significant inhibition.."

The references appear to be formatted ok and figures are clear for the most part, although Panel 9E has no units.

Referee #2:

General comments:

The aim of the study by Thomasi et al. was to evaluate the contribution of S100B to glia mediated regulation of enteric neural circuitry and motor activity. The authors used organ bath experiments to record CMCs and performed imaging studies in mice with cell-specific expression of GCaMP and tdTomato to evaluate how manipulations to S100B modulate glial and neuronal activity. Immunohistochemical studies were also undertaken.

The rationale for the study is well thought out and the experiments that have been performed to test the hypothesis are appropriate. The methods and analyses used are sophisticated, particularly for correlation studies. The manuscript is beautifully written, and the findings are discussed well with inclusion of appropriate references. No major flaws are noted. The findings of this study improve our understanding of how S100B and glia modulate cholinergic neuronal activity in the gut. Separation of the data to evaluate sex-specific differences is also a strength of the study and provides greater insights into how glial and neuronal activity is modulated differently between sexes.

However, there are some fairly minor concerns that need to be addressed to improve the overall quality of this manuscript. These are outlined in the specific comments below.

Specific comments:

- p.9-10, lines 247-251. Why were different concentrations of arundic acid (50 and 300 uM) and pentamidine (10 and 300uM) used for Ca²⁺ imaging vs. recording of CMCs? Why was such a higher concentration needed for CMCs? Do the authors have any concern about potential off-target effects at this concentration?
- Figure 2A, lower panel. A description for this is missing from the figure legend/panel. Do all drugs/antibody wash out equally as well?
- Figure 8 and p.25, lines 581-582. Though activity in neurons 1 and 2 appears to mostly recover, there doesn't appear to be much recovery in neurons 3 and 4 (neuron 3 especially). Is this typical in neurons with these firing patterns or just in this particular example?
- The effects on cholinergic neurons using Chat mice are convincing. As mentioned by the authors on p.32, lines 763-765 Wnt1+ neurons include both excitatory and inhibitory populations. Do the authors have any idea on how S100B may affect nitrergic (or other) neurons? Perhaps they could speculate if not yet known.
- The authors mention the proposed mechanism of cyclic release of tonic neuronal inhibition of ICC to CMCs (lines 87-88). Perhaps the authors could speculate regarding how glia (directly or indirectly) may influence ICC activity.

Minor points:

- p.25, line 582. The word "neurons" is missing after ChAT(+).
- p.33, line 769. Should be Cx43 rather than C43.
- p.34, line 811. Misspelling of "underlie" (says underly).
- References, Debnath et al. 2025 is duplicated. Referred to as 2025a in first citation (line 407) and 2025b subsequently (lines 787, 792).

Overall, this is an elegant study that has approached the topic in a novel and innovative fashion. The conclusions presented are valid based on the experimental evidence provided. This study adds to our knowledge and understanding of how S100B from enteric glia regulates gut motor neurocircuits.

END OF COMMENTS

Response to Referees and Editors (blue letters in a gray background)

We thank the reviewers and editors for their careful evaluation of our work. Both reviewers provided excellent feedback on how to improve our work, and we have addressed each in full with changes to the manuscript text, figures, and in the following point-by-point response. We feel that these changes have improved the quality and clarity of our study and hope that the reviewers agree that it is now acceptable for publication.

EDITOR COMMENTS

Reviewing Editor:

Thank you for submitting your work to the Journal.

The two referees that have reviewed the paper both agree in general that the study is well performed, experimental approach is sophisticated, and data analysis and separation of data based on sex are both highly commendable features of the paper.

An important concern is that reviewer 1 points to an earlier study in the Journal (<https://pubmed.ncbi.nlm.nih.gov/22063626/>) which previously demonstrated glial cells interact with the ENS during colonic motor complexes (CMCs), and makes some of the same observations as the current study. The authors should cite this previous paper noted by reviewer 1 (<https://pubmed.ncbi.nlm.nih.gov/22063626/>) and explicitly provide evidence on how the current study expands on or differentiates itself from this previous work.

Reply: We would like to begin by thanking the editors for their support of our work and for their excellent feedback. We have addressed each point raised by the reviewers in full and hope that you now find the manuscript acceptable for publication. With regard to the concern of Reviewer 1 questioning how our study was different from prior work that showed enteric glial activity during CMCs, we would like to emphasize that our study investigated the downstream effects resulting from glial cell activity and are fundamentally different from what was shown in prior work. The excellent work by Broadhead et al showed that enteric glial cells exhibit Ca^{2+} responses during CMCs in isolated preparations, but this work did not address if or how enteric glial cells exerted reciprocal effects on neurons or CMC behaviors. This is where our work is unique. Here, we show that S100B is a gliotransmitter that exerts an inhibitory tone over myenteric neurocircuits that is permissive for CMC behaviors. This is a major advance beyond simply observing that glial cell activity is associated with CMCs. We have revised the text to clarify this difference and to describe how our findings build upon the prior work

by Broadhead et al. to provide a new understanding of enteric glia function, CMCs, and the regulation of spontaneous activity within the enteric nervous system.

In addition, in agreement with reviewer 2, a discussion on how S100B signalling from glia might interact with other cell types would be most welcome. In the submitted paper, a clear emphasis is put on s100B effects on cholinergic neurons (for reasons the authors explain). However, activation of nNOS+ neurons are known to influence CMCs, which still occur in the presence of atropine (<https://pubmed.ncbi.nlm.nih.gov/35446689/>) through signalling via ICC. While any such discussions on this would be speculative, a brief discussion would be beneficial.

Reply: This is an excellent suggestion, and we have added the following text to the discussion to consider these potential interactions. We agree to consider this aspect to be interesting and will be beneficial for our paper. Following, the added text and references:

“Therefore, a further understanding of how glial S100B regulates neuron-to-neuron interactions and intrinsic inputs that culminate in the rhythmic properties of the colon will be important for a better understanding of gut health and disease. This may include gliotransmission effects into other subpopulations that are fundamental to CMC, such as nitrergic neurons. Specialized interactions between enteric glia and nitrergic neurons have been proposed for tonic inhibition and CMC generation periods (Smith & Koh, 2017), and may play a key role in pathological conditions where glial purinergic signaling contributes to nitrergic neuron death (Brown *et al.*, 2016). In the same vein, glial signaling may critically influence ICCs and the SIP syncytium by their purinergic signaling (Gulbransen & Sharkey, 2009a; Montalant *et al.*, 2021; Hendler *et al.*, 2024). Indirect effects on the smooth muscle environment may also occur through S100B glial modulation of enteric neurons, notably nitrergic subpopulation. In the CNS, S100B promotes neuronal death by astrocyte release of nitric oxide (Hu *et al.*, 1997). Similarly, S100B-RAGE axis activation induces nitric oxide production (Villarreal *et al.*, 2014) which may alter the balance of enteric neurotransmission and intercellular interactions, ultimately affecting motility outputs.”

Please also take careful note of the Journals statistics policy in any resubmission, as some summary graphs show exact p values for statistical significance and some do not. Refer here: https://jp.msubmit.net/cgi-bin/main.plex?form_type=display_requirements#statistics

Reply: Thank you, a careful revision on this matter was performed to be in accordance with the Journal guidelines.

Also note that the exact anaesthetic protocol used for animal sacrifice must be stated in the Methods.

Reply: Protocols for animal sacrifice are fully described in the Methods section and follow the NIH Guidelines, the Institutional Animal Care and Use Committee at Michigan State University and the ARRIVE guidelines 2.0 adopted by the journal. As described in the following text excerpt from the Methods section, cervical dislocation and decapitation was used to avoid complications of anesthesia on ENS activity: "Experiments were conducted with male and female mice, 8-12 weeks old. Animal euthanasia was conducted by trained personnel using cervical dislocation with a second step of decapitation. Animal death was confirmed by observation of heartbeat cessation."

Referee #1

1. My major concern with this study is that enteric glia have already been demonstrated to be involved in the motor behaviour of the colon of mice, called CMCs, see 10.1113/jphysiol.2011.219519. Hence, the novelty of the study is somewhat in doubt.

Reply: We agree that the study by Broadhead et al. was excellent, groundbreaking work and we apologize for our oversight in referencing this study. This reference was mistakenly deleted during multiple rounds of revisions before submission. Indeed, Broadhead et al. showed that enteric glial cells exhibit Ca^{2+} responses during CMCs in whole mounts of myenteric plexus in vitro. However, the relevance of glial cell activity in CMC control has remained in question. Our work offers new insight by showing the consequence of a transmitter released by glia (S100B, a gliotransmitter) on CMCs through effects on neuron subtypes in gut motor neurocircuits. Thus, we show that glial cells are not simply detectors of neuron activity during CMCs but are active effectors that exert reciprocal control over CMC neurocircuits. This observation provides a critical answer to the work begun by Broadhead and shows that glial effector systems involve the transmitter S100B. To clarify our novelty, we have revised the manuscript text to add the study by Broadhead et al. and have emphasized how our findings contribute to a new understanding of enteric glia function, CMCs, and the regulation of spontaneous activity within the enteric nervous system.

2. How do the authors know that arundic acid doesn't act on myenteric neurons?

Reply: Thank you for allowing us to clarify this important point. Previous studies investigating the effects of arundic acid (also known as ONO-2506) have included co-culture systems of astrocytes and neurons. These studies reported that arundic acid did not alter neuronal viability or growth but significantly reduced S100B mRNA levels in glial cells (PMID: 12045671). When neuronal cultures were exposed to glutamate-induced toxicity, arundic acid did not confer protection in neuron-only cultures, but neuroprotection was observed in the presence of astrocytes, suggesting an indirect, glia-mediated mechanism (PMID: 12045671). A substantial body of literature supports the role of arundic acid in modulating S100B (a protein confined to enteric glial cells in the myenteric plexus) and in providing protection against neuroinflammation (PMID: 32918255), neuronal death in ischemic models (PMID: 12045671), and neurodegeneration (PMID: 15567338), likely through S100B downregulation. Therefore, we interpret our findings in the context of glial regulation of neuronal function via S100B modulation. Importantly, in *ex vivo* models of the developing enteric nervous system, arundic acid treatment did not affect the density of HuC/D α neurons (PMID: 28280459). However, detailed assessments of arundic acid's direct effects on enteric neuronal physiology (such as gene expression, enzymatic activity, or proteomics) have not yet been reported. Thus, while our data support a predominantly glial-mediated mechanism, we acknowledge that direct neuronal effects cannot be entirely ruled out. We have addressed this point in the revised Discussion section of the manuscript:

“Therefore, effects of S100B on enteric neuronal activity could conceivably occur subsequent to effects on glial excitability and gliotransmission or direct effects on neurotransmission that involve extracellular Ca²⁺ buffering. Direct effects of arundic acid on neurons have not been reported in studies of neurons in cell culture or ENS development (Tateishi *et al.*, 2002; Asano *et al.*, 2005; Hao *et al.*, 2017); however, we cannot completely rule out the possibility that some of the effects we observed were due to direct actions on neurons. Given the similar effects of multiple drugs and antibody sequestering strategies on CMC function and ENS cell excitability, we believe our data are most consistent with glial regulation, and this would agree with prior data showing that S100B affects neurotransmission in the brain by regulating [Ca²⁺]_e (Morquette *et al.*, 2015a).”

3. The mechanism of action of pentamidine is not well understood. Without a clear understanding of how the drug works doesn't provide strong support that the drug is specific of enteric glia.

Reply: Thank you for allowing us to clarify the selectivity of this drug. Pentamidine is a well-established antimicrobial compound (PMID: 33317111; PMID: 35378227), but it

also acts as a direct cellular regulator of the Receptor for Advanced Glycation Endproducts (RAGE), interfering with ligand-receptor interactions. Specifically, it has been shown to modulate the binding of S100B to RAGE (PMID: 18331229; PMID: 18602402), as well as to directly influence calcium Ca^{2+} binding to S100B (PMID: 23256816). In our study, pentamidine was selected as a direct modulator of the S100B-RAGE signaling pathway to investigate its contribution to physiological ENS excitability, potentially involving both neuronal and glial receptors. However, considering the low expression of RAGE in the healthy myenteric plexus and the absence of changes in ENS calcium activity following RAGE antibody application, we excluded the RAGE pathway as a major contributor to our findings. Instead, the similar effects observed with pentamidine, and S100B-blocking antibodies support the interpretation that our results reflect extracellular modulation of S100B- Ca^{2+} interactions. This interpretation has been addressed and expanded upon in the revised Results section:

“S100B mainly exerts its actions in the extracellular space by binding Ca^{2+} or by engaging Receptors for Advanced Glycation End-products (RAGE) (Barger & Eldik, 1992; Zimmer & Weber, 2010; Villarreal *et al.*, 2011; Hagemeyer *et al.*, 2018). Pentamidine is a drug with broad actions on S100B-RAGE axis both by directly disrupting Ca^{2+} /p53 binding site of S100B and limiting its ability to bind Ca^{2+} to the glial protein, or by blocking S100B ligand activity on RAGE. (Charpentier *et al.*, 2008; Hartman *et al.*, 2013).”

“The effects observed with pentamidine could result from how this drug interferes with RAGE ligands, interactions between S100B and Ca^{2+} or by directly blocking interactions between S100B and RAGE. To differentiate between these possibilities, we repeated CMC recordings and recordings of spontaneous ENS activity in the presence of the high-affinity RAGE antagonist FPS ZM1 (1 μ M) or RAGE antibodies, respectively (Deane *et al.*, 2012). In contrast with the previous manipulations of S100B, FPS ZM1 did not affect CMC behavior (**Fig 6A-C**). This observation was supported by immunolabeling experiments that showed an absence of RAGE expression in the myenteric plexus of the healthy colon (**Fig 6D**). This supports the idea that pentamidine effects were due mainly to changes in S100B- Ca^{2+} interactions. Ca^{2+} imaging experiments also showed no effects on enteric neuron excitability following incubation with anti-RAGE antibodies (**Fig 6E-F**). Therefore, pentamidine most likely exerts its effects on ENS activity in healthy myenteric plexus by Ca^{2+} regulation in the extracellular space.”

4. Figure 1 of reference <https://pmc.ncbi.nlm.nih.gov/articles/PMC5322270/> it has been shown that S100B expression colocalises with enteric neural crest cells that are SOX10 positive. Hence, the idea that S100B is exclusive to enteric glia and not expressed in the ENS is not unequivocal <https://doi.org/10.3389/fncel.2017.00042> S100B is expressed in some neurons of rat see (10.1016/0306-4522(94)00615-c), and in mice

SB100 expression has been shown in astrocytes and neurons in specific parts of the brain see: <https://doi.org/10.1523/JNEUROSCI.12-11-04337.1992> The authors need to validate that S100B is not expressed in enteric neurons of mouse colon, using more than antibodies?

Reply: Thank you for raising this important point. The developmental data referred to by the reviewer reflects a time point when progenitor cells express many proteins that are not sustained in mature ENS cells. All ENS progenitors express SOX10 but those that commit to a neural fate downregulate SOX10 and those that become glia maintain it. It is important to note that in the figure referred to by the reviewer that S100B is restricted to SOX10+ glia at P0. By P0, enteric neurons have lost expression of SOX10 and this marker is only maintained in glia. Therefore, the data referred to by the reviewer reinforce the concept that S100B is restricted to enteric glia in the myenteric plexus. Multiple lines of evidence support the conclusion that S100B is restricted to enteric glia in the mature myenteric plexus (PMID: 7043279, PMID: 25161129, PMID: 24155689). For example, transgenic mice expressing green fluorescent protein driven by the glial specific S100B promoter show clear glial expression with no expression observed in neurons (PMID: 19250649). In addition, bulk RNA-seq and single-nucleus RNA-seq data from FACS isolated neuronal and glial nuclei showed that S100B is among the most highly enriched genes in the glial fraction and one of the top glia-enriched markers overall. Single-cell multi-ome sequencing of adult ENS also demonstrated strong selectivity of S100B for glial pools and not for enteric neurons (PMID: 33288908, PMID: 36857184, PMID: 33010250). scRNAseq evaluation of visceral smooth muscle cells also did not show significant S100B expression.

Collectively, our data and conclusions are in line with the multiple supporting lines of evidence from prior studies that show S100B is restricted to enteric glia in the mature myenteric plexus. To reinforce this feature and strength the relevance of glial S100B from murine studies to human physiology, we added full thickness human colon cross-section labeled with PGP9.5 (neurons) and S100B (glia) to Figure 1. We see S100B labeled in enteric glia (like in mice experiments) while PGP9.5 displays neuronal cell bodies and fibers. We added a short description that is found below:

“To support the translational relevance of murine ENS studies to human physiology, PGP9.5 and S100B were labeled in full thickness human colon cross-sections. The reactions reveal the organization of colon layers and ENS with PGP9.5 marking neuronal cell bodies and fibers, and S100B confined to surrounding enteric glia (**Fig 1G-H**). This spatial organization is consistent with that observed in mouse models and demonstrates S100B specificity to the glial type. “

5. Line 113 states: based on the current data, this statement should be softened: "Enteric glia are unique neuroglia of the ENS and the only cellular source of S100B within gut motor neurocircuits." Again, what is the evidence no other cells express S100B in these preparations of gut?

Reply: Thank you for pointing out this statement. As we mentioned in the previous comment, there are multiple lines of evidence showing that S100B is specific for enteric glia in the myenteric plexus (see refs described above in point 4). The neurocircuitry that controls gut motor functions is housed in the myenteric plexus. This layer of the gut contains glia and neurons that are surrounded by smooth muscle cells and muscularis macrophages that populate peri- and intraganglionic structures. While it may be possible that smooth muscle cells or macrophages upregulate S100B expression in disease (e.g. PMID: 28693920), this does not occur in healthy systems as assessed in this study. In this context, S100B is confined to enteric glia within gut motor neurocircuits.

6. A major reference by Broadhead M (2012) Journal of Physiology is not cited, that studied role of glia in CMCs. This should be quoted.

Reply: Thank you for catching this mistake. This reference was included in prior versions of the manuscript but was mistakenly deleted during revisions prior to submission. We did also discuss the data of "Broadhead et al., 2012" by referencing the review paper of "Broadhead and Miles 2021" (PMID: 33894330), which is another fundamental reference for our paper. However, we have fixed this issue by correctly adding the reference and revising the introduction section.

"Crosstalk between enteric neurons and glia is extensive and glial signaling mechanisms enacted in response to synaptic transmission function to modulate enteric neural circuitry underlying gut motility (Gulbransen & Sharkey, 2009a; Boesmans *et al.*, 2013a, 2013b; Delvalle *et al.*, 2018a; Ahmadzai *et al.*, 2021a; Seguella *et al.*, 2022). **Moreover, enteric glia are active during CMCs and their activity is associated with activity in the surrounding neurons and varicosities (Broadhead et al., 2012).** Yet, whether enteric glial signaling processes control enteric CPG behaviors as astrocytes do in the brain remains unknown. Given the similar functions of enteric glia and astrocytes in neural networks, **the occurrence of neuron-glia crosstalk during CMCs (Broadhead et al., 2012),** and that enteric glia are rich in S100B, we reasoned that S100B is a likely candidate by which enteric glia modulate the behavior of enteric CPG neurocircuits underlying CMC behaviors in the gut"

7. Figure legend 9 is slightly confusing. The title states "Impairing S100B release

disrupts correlation and synchronization patterns among cholinergic neurons in the myenteric plexus." But figure 9B legend states that in "arundic acid increases correlated (green) and highly correlated (yellow) neuron pairs." Figure 9E shows more yellow which to me means increased correlation of neuronal pairs. Please clarify what this is showing.

Reply: We apologize for the lack of clarity in this caption. As a context: following S100B inhibition, we identified an increase in correlations, which grouped neurons into less diverse clusters (because they are strongly correlated with similar patterns of response). In this case, this modulation points towards a dysfunctional pattern. We included a detailed discussion on this matter, connecting enhanced cell activity correlations or synchronization to disease states. To clarify this point, we replaced the word "disrupts" with "modulates" since different trends of the cell profile and interactions are described in this Figure.

Line 374: rather than " caused a major failure of CMC activity", may I suggest "caused a significant inhibition.."

Reply: Thank you for your suggestion. We decided to keep the description as it is because we are not sure that there was an inhibitory mechanism involved in the function abolishment. CMCs may fail due to several mechanisms that could include inhibition, a lack of synchronization cues, or heightened activity in certain neuron subsets. Therefore, we feel that "failure" is more appropriate than "inhibition" in this case.

The references appear to be formatted ok and figures are clear for the most part, although Panel 9E has no units.

Reply: Thank you for catching this mistake in Figure 9 (and 10). This has been corrected in the revised figures.

Referee #2:

The aim of the study by Thomasi et al. was to evaluate the contribution of S100B to glia mediated regulation of enteric neural circuitry and motor activity. The authors used organ bath experiments to record CMCs and performed imaging studies in mice with cell-specific expression of GCaMP and tdTomato to evaluate how manipulations to S100B modulate glial and neuronal activity. Immunohistochemical studies were also undertaken.

The rationale for the study is well thought out and the experiments that have been performed to test the hypothesis are appropriate. The methods and analyses used are sophisticated, particularly for correlation studies. The manuscript is beautifully written, and the findings are discussed well with inclusion of appropriate references. No major flaws are noted. The findings of this study improve our understanding of how S100B and glia modulate cholinergic neuronal activity in the gut. Separation of the data to evaluate sex-specific differences is also a strength of the study and provides greater insights into how glial and neuronal activity is modulated differently between sexes.

However, there are some fairly minor concerns that need to be addressed to improve the overall quality of this manuscript. These are outlined in the specific comments below.

Reply: We thank the reviewer for their positive evaluation of our work and for their excellent comments. We have considered each carefully and have addressed them as described below in the point-by-point response.

Specific comments:

- p.9-10, lines 247-251. Why were different concentrations of arundic acid (50 and 300 uM) and pentamidine (10 and 300uM) used for Ca²⁺ imaging vs. recording of CMCs? Why was such a higher concentration needed for CMCs? Do the authors have any concern about potential off-target effects at this concentration?

Reply: Thank you for your inquiry. The two experiments involved distinct sample preparations and media exposure conditions. The organ bath assays used intact colonic segments placed in large-volume chambers. In this setup, any substance applied to the medium must diffuse through the serosa and longitudinal muscle layers to reach the myenteric plexus. Therefore, higher drug concentrations were used in these experiments, as supported by previous studies using intact organs (PMID: 28280459; PMID: 26693173). For the live cell imaging experiments, the longitudinal muscle was dissected to directly expose the myenteric plexus to the medium. In these preparations, drug concentrations were selected based on established *ex vivo* and *in vitro* studies. While the concentrations differ between experimental approaches, all values used in our study are supported by the literature (PMID: 33727126; PMID: 17678654; PMID: 28280459; PMID: 26137161; PMID: 26295040). We acknowledge that off-target effects are always a consideration when using pharmacological agents. However, the concentrations chosen in our work fall within ranges reported to be effective while minimizing nonspecific actions.

- Figure 2A, lower panel. A description for this is missing from the figure legend/panel. Do all drugs/antibody wash out equally as well?

Reply: Thank you for calling attention to this matter. The caption was updated to include the reference to arundic acid, the drug washed out in that assay. Furthermore, organ baths washouts following pentamidine application did not show recovery and antibodies were not tested. Since we did not find changes following FPS-ZM1 application, no washouts were conducted.

Below, the texts that were updated:

“Arundic acid decreased the amplitude of oral and aboral contractions by 70% and 84% (**Fig. 2B**), respectively, and the integral of contractions by 82% and 89% (**Fig. 2C**). Notably, these effects were reversible, and CMC activity resumed after **arundic acid** washout and return to normal media (**Fig 2A, bottom**).”

Caption: “**(A)** Representative traces of colonic motor complex (CMC) behavior in the mouse colon under control conditions and following treatment with drugs to manipulate S100B. Solid lines show motor responses at oral recording sites while dashed lines show motor corresponding responses at aboral recording sites in the same organ. On the bottom, washout CMC **following arundic acid** washout”

- Figure 8 and p.25, lines 581-582. Though activity in neurons 1 and 2 appears to mostly recover, there doesn't appear to be much recovery in neurons 3 and 4 (neuron 3

especially). Is this typical in neurons with these firing patterns or just in this particular example?

Reply: That is an excellent question. The behavior of the particular neurons shown in this example illustrates some of the profiles that we observed. While some neurons did not return to the previous pattern of firing, others from the same cluster did. Some neurons even kept the same profile but decreased amplitudes. This can be partially visualized in the clusters before and after arundic acid incubation in figure 9. While the current data analysis was sufficient to identify neurons that recovered or not, the criteria that define this feature are complex and were not part of this study. However, this is something that we plan to assess in future work (e.g. the neurochemical/functional/pathway specificity of that neurons that recover or not).

- The effects on cholinergic neurons using Chat mice are convincing. As mentioned by the authors on p.32, lines 763-765 Wnt1+ neurons include both excitatory and inhibitory populations. Do the authors have any idea on how S100B may affect nitrenergic (or other) neurons? Perhaps they could speculate if not yet known.

Reply: This is an excellent point, and we are, in fact, conducting related studies to understand how glia affect different subtypes of enteric neurons. In the current work, we chose to focus on cholinergic neurons given their central role in coordinating CMC activity in the ENS. However, we completely agree that studying potential effects of S100B on other subtypes of enteric neurons would be extremely valuable in future work. How glia affect nitrenergic neurons is mostly unknown but specialized interactions between nitrenergic neurons and enteric glia were proposed by Terrance Smith (PMID: 27789457) and our prior studies show that glia are involved in nitrenergic neuron death during inflammation (PMID: 26771001). It is also known that enteric glia detect and modify specific synaptic pathways in the ENS (PMID: 34593632) and that enteric glia release mediators (PGE2 and GABA) that suppress inhibitory pathways (PMID: 28838986, PMID: 37988454). Therefore, it is certainly possible that enteric glia exert modulatory effects over nitrenergic neurons. While there is no evidence for enteric glial S100B actions over nitrenergic neurons during physiological conditions, there are potential effects during inflammation and disease. For example, in the CNS, S100B promotes neuronal death by astrocyte release of nitric oxide (PMID: 9375660). Similarly, S100B could act through RAGE pathways in disease (PMID: 24923428) to induce nitric oxide production and change balance of enteric neurotransmission. We appreciate the question and part of this discussion was included in the revised text:

“Therefore, a further understanding of how glial S100B regulates neuron-to-neuron interactions and intrinsic inputs that culminate in the rhythmic properties of the colon will be important for a better understanding of gut health and disease. This may include

gliotransmission effects into other subpopulations that are fundamental to CMC, such as nitrergic neurons. Specialized interactions between enteric glia and nitrergic neurons have been proposed for tonic inhibition and CMC generation periods (Smith & Koh, 2017), and may play a key role in pathological conditions where glial purinergic signaling contributes to nitrergic neuron death (Brown *et al.*, 2016)”

- The authors mention the proposed mechanism of cyclic release of tonic neuronal inhibition of ICC to CMCs (lines 87-88). Perhaps the authors could speculate regarding how glia (directly or indirectly) may influence ICC activity.

Reply: Potential direct effects of glia on ICC activity are interesting concepts. While there is no work addressing this possibility, it is possible that enteric glia could influence ICC activity in a number of ways. First, enteric glia could affect ICC activity either directly or indirectly by releasing purines. Purinergic transmission plays important roles in similar CPG circuits and enteric glia are active in purinergic transmission in the ENS (PMID: 34558208, PMID: 19250649, PMID: 39612055. Glial purines could act directly on ICCs or could influence ICC activity through effects on neurons. Second, our data show that S100B controls ENS activity, which could include changes to nitrergic neurons. This would ultimately modify tonic inhibition in myenteric neurocircuits and change their interactions with ICCs that control CMCs (PMID: 35446689). Lastly, the effects of enteric glia may change in conditions that promote neuroplasticity such as disease and inflammation. In these circumstances, upregulation of RAGE pathways could expand the actions of S100B and contribute to alterations in neurons excitability with subsequent effects on ICCs and the SIP syncytium.

Those references and discussion were added to our Discussion session:

“Therefore, a further understanding of how glial S100B regulates neuron-to-neuron interactions and intrinsic inputs that culminate in the rhythmic properties of the colon will be important for a better understanding of gut health and disease. This may include gliotransmission effects into other subpopulations that are fundamental to CMC, such as nitrergic neurons. Specialized interactions between enteric glia and nitrergic neurons have been proposed for tonic inhibition and CMC generation periods (Smith & Koh, 2017), and may play a key role in pathological conditions where glial purinergic signaling contributes to nitrergic neuron death (Brown *et al.*, 2016). In the same vein, glial signaling may critically influence ICCs and the SIP syncytium by their purinergic signaling (Gulbransen & Sharkey, 2009a; Montalant *et al.*, 2021; Hendler *et al.*, 2024). Indirect effects on the smooth muscle environment may also occur through S100B glial modulation of enteric neurons, notably nitrergic subpopulation. In the CNS, S100B promotes neuronal death by astrocyte release of nitric oxide (Hu *et al.*, 1997). Similarly, S100B-RAGE axis activation induces nitric oxide production (Villarreal *et al.*, 2014)

which may alter the balance of enteric neurotransmission and intercellular interactions, ultimately affecting motility outputs.”

Minor points:

- p.25, line 582. The word "neurons" is missing after ChAT(+).

Reply: Thank you, the sentence was corrected.

- p.33, line 769. Should be Cx43 rather than C43.

Reply: Thank you, the sentence was corrected.

- p.34, line 811. Misspelling of "underlie" (says underly).

Reply: Thank you, the sentence was corrected.

- References, Debnath et al. 2025 is duplicated. Referred to as 2025a in first citation (line 407) and 2025b subsequently (lines 787, 792).

Reply: Thank you, reference list was correctly updated to include this paper as a unique reference paper.

Overall, this is an elegant study that has approached the topic in a novel and innovative fashion. The conclusions presented are valid based on the experimental evidence provided. This study adds to our knowledge and understanding of how S100B from enteric glia regulates gut motor neurocircuits.

Dear Dr Thomasi,

Re: JP-RP-2025-289410R1 "Enteric glial S100B controls rhythmic colonic functions by regulating excitability and specificity in gut motor neurocircuits" by Beatriz Thomasi, Rafaella Lavalle, Jonathon L. McClain, Julia R. Jamka, Luisa Seguella, and Brian David Gulbransen

We are pleased to tell you that your paper has been accepted for publication in The Journal of Physiology.

Yours sincerely,

Kim Barrett
Senior Editor
The Journal of Physiology

If you would like to receive our 'Research Roundup', a monthly newsletter highlighting the cutting-edge research published in The Physiological Society's family of journals (The Journal of Physiology, Experimental Physiology, Physiological Reports, The Journal of Nutritional Physiology and The Journal of Precision Medicine: Health and Disease), please click this link, fill in your name and email address and select 'Research Roundup':
<https://www.physoc.org/journals-and-media/membernews>

- You can help your research get the attention it deserves! Check out Wiley's free Promotion Guide for best-practice recommendations for promoting your work at: www.wileyauthors.com/eeo/guide. You can learn more about Wiley Editing Services which offers professional video, design, and writing services to create shareable video abstracts, infographics, conference posters, lay summaries, and research news stories for your research at: www.wileyauthors.com/eeo/promotion.

EDITOR COMMENTS

Reviewing Editor:

Thank you for your diligent and considered response to the comments and suggestions by the reviewers.

The impact of this study has been enhanced by incorporating the new revisions.

REFEREE COMMENTS

Referee #2:

The authors have carefully revised the manuscript to address the reviewer comments. This reviewer has no further comments to add. Congratulations to the authors on a great study!